# New-to-nature $CO_2$-dependent acetyl-CoA assimilation enabled by an engineered $B_{12}$-dependent acyl-CoA mutase

Helena Schulz-Mirbach [1,4], Philipp Wichmann [1,4], Ari Satanowski [1,2,4] ✉,
Helen Meusel[2], Tong Wu [2], Maren Nattermann[1], Simon Burgener [1],
Nicole Paczia [1], Arren Bar-Even [2,5] & Tobias J. Erb [1,3] ✉

Acetyl-CoA is a key metabolic intermediate and the product of various natural and synthetic one-carbon (C1) assimilation pathways. While an efficient conversion of acetyl-CoA into other central metabolites, such as pyruvate, is imperative for high biomass yields, available aerobic pathways typically release previously fixed carbon in the form of $CO_2$. To overcome this loss of carbon, we develop a new-to-nature pathway, the Lcm module, in this study. The Lcm module provides a direct link between acetyl-CoA and pyruvate, is shorter than any other oxygen-tolerant route and notably fixes $CO_2$, instead of releasing it. The Lcm module relies on the new-to-nature activity of a coenzyme $B_{12}$-dependent mutase for the conversion of 3-hydroxypropionyl-CoA into lactyl-CoA. We demonstrate Lcm activity of the scaffold enzyme 2-hydroxyisobutyryl-CoA mutase from *Bacillus massiliosenegalensis*, and further improve catalytic efficiency 10-fold by combining in vivo targeted hypermutation and adaptive evolution in an engineered *Escherichia coli* selection strain. Finally, in a proof-of-principle, we demonstrate the complete Lcm module in vitro. Overall, our work demonstrates a synthetic $CO_2$-incorporating acetyl-CoA assimilation route that expands the metabolic solution space of central carbon metabolism, providing options for synthetic biology and metabolic engineering.

Microbial one-carbon (C1) assimilation plays a key role in the global carbon cycle[1–3], where it is responsible for the fixation of inorganic C1-compounds, such as $CO_2$, into biomass. Microbial C1-metabolism is also of increasing interest for biotechnology: employing natural and engineered microbial C1-metabolism is considered essential for building a sustainable future biotechnology that is based on $CO_2$ as the carbon source[4]. Almost all natural pathways for autotrophic carbon fixation yield acetyl-CoA as the primary fixation product, including the reductive acetyl-CoA pathway as the most ancient and prominent example[5–7] (Supplementary Table 1). Furthermore, acetyl-CoA is the product of several proposed new-to-nature C1-fixation pathways, for example, the THETA cycle[8,9], the 2-hydroxyglutarate-reductive TCA cycle[8], the reductive acetyl-CoA bicycle[10,11] as well as the serine-threonine cycle[12].

Once formed, the central hub metabolite acetyl-CoA needs to be further converted (i.e., assimilated) into all other cellular building blocks[5]. Multiple acetyl-CoA assimilation pathways have evolved in diverse organisms, the most widespread are the glyoxylate cycle and

[1]Max Planck Institute for Terrestrial Microbiology, Karl-von-Frisch-Str. 10, Marburg, Germany. [2]Max Planck Institute of Molecular Plant Physiology, Am Mühlenberg 1, Potsdam-Golm, Germany. [3]Center for Synthetic Microbiology (SYNMIKRO), Karl-von-Frisch-Straße 14, Marburg, Germany. [4]These authors contributed equally: Helena Schulz-Mirbach, Philipp Wichmann, Ari Satanowski. [5]Deceased: Arren Bar-Even. ✉e-mail: ari.satanowski@gmail.com; toerb@mpi-marburg.mpg.de

the ethylmalonyl-CoA pathway[13,14]. Additional routes have been identified or proposed, such as the methylaspartate cycle in halophilic archaea[15] and the citramalate cycle in proteobacteria[16]. Notably, all of the above-mentioned acetyl-CoA assimilation pathways ultimately lead to the formation of a C4 compound, in most cases malate, which is subsequently converted into oxaloacetate (Supplementary Fig. 1). However, to synthesize C3 intermediates of central carbon metabolism, such as pyruvate or phosphoenolpyruvate (PEP), oxaloacetate needs to be decarboxylated, which leads to a loss of previously fixed carbon. In the context of microbial $CO_2$ valorization, such loss of carbon is not desired, as it requires additional investment of energy and other cellular resources for (re-)assimilation of the released $CO_2$.

Assimilation of acetyl-CoA can also proceed without the release of $CO_2$ via *reductive* routes. These reductive routes involve a net investment of reducing power and co-assimilation of $CO_2$, further contributing to the cell's overall C1-fixation (Supplementary Fig. 2). For instance, many anaerobic organisms utilize pyruvate synthase (pyruvate-ferredoxin oxidoreductase), an oxygen-sensitive enzyme, for the direct reductive carboxylation of acetyl-CoA to pyruvate. This route is used for example by microbes operating the reductive acetyl-CoA pathway, reductive TCA cycle or as part of the archaeal dicarboxylate/4-hydroxybutyrate cycle[17,18]. Reductive (co-)assimilation of acetyl-CoA with $CO_2$ is also part of the oxygen-tolerant reaction sequences of the 3-hydroxypropionate bicycle[19] or the ethylmalonyl-CoA pathway[20–22]. However, the latter two routes are very long (>10 enzymes,

Supplementary Fig. 2) and still yield a C4 compound that is decarboxylated again to form gluconeogenic C3 compounds.

In this work, we present the Lcm module, an oxygen-tolerant, $CO_2$-incorporating acetyl-CoA assimilation pathway that provides a link from acetyl-CoA to pyruvate and other C3 intermediates of gluconeogenesis (Fig. 1). This pathway is based on the carboxylation and conversion of acetyl-CoA into the C3 compound 3-hydroxypropionyl-CoA (3-HP-CoA), followed by its conversion into lactyl-CoA, and ultimately pyruvate. Key to this pathway is a new-to-nature carbon rearrangement from 3-HP-CoA into lactyl-CoA (lactyl-CoA mutase, Lcm), which we establish by leveraging the natural promiscuity of $B_{12}$-dependent mutases. We reconstitute the cascade from acetyl-CoA to lactyl-CoA in vitro and identify Lcm as the limiting bottleneck. Establishing the Lcm reaction in a selection strain allows us to combine in vivo random mutagenesis and adaptive evolution to further improve Lcm. We confirm improved Lcm activity in vitro and report up to 10-fold improved catalytic efficiency of the identified Lcm variants compared to the wild-type enzyme. Overall, this work lays the foundation for a new-to-nature acetyl-CoA assimilation module that directly links acetyl-CoA with pyruvate. This module could improve acetyl-CoA assimilation in the context of several C1-assimilating pathways that were recently demonstrated[9,12], and also opens the way for the implementation of highly efficient synthetic $CO_2$ fixation cycles in the future (Supplementary Fig. 3)[11].

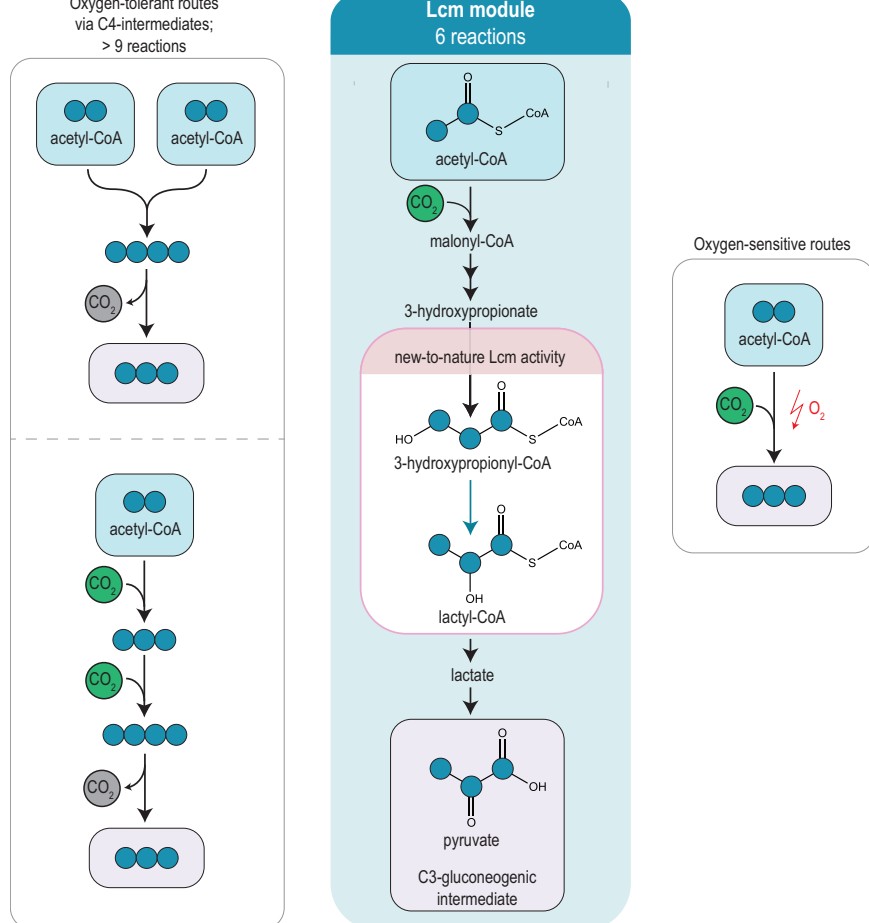

**Fig. 1 | The Lcm module.** The Lcm module is a short route (six reactions) that enables the $CO_2$-incorporating conversion of acetyl-CoA into pyruvate via a new-to-nature lactyl-CoA mutase (Lcm). Naturally existing (or synthetic) metabolic routes for this conversion either involve C4 intermediates, require more than ten steps and a decarboxylation for pyruvate synthesis (left box), or are oxygen-sensitive (right box). All pathways are shown in more detail in Supplementary Figs. 1 and 2. Closed circles indicate carbon atoms, assimilated $CO_2$ is shown in green and released $CO_2$ is shown in gray.

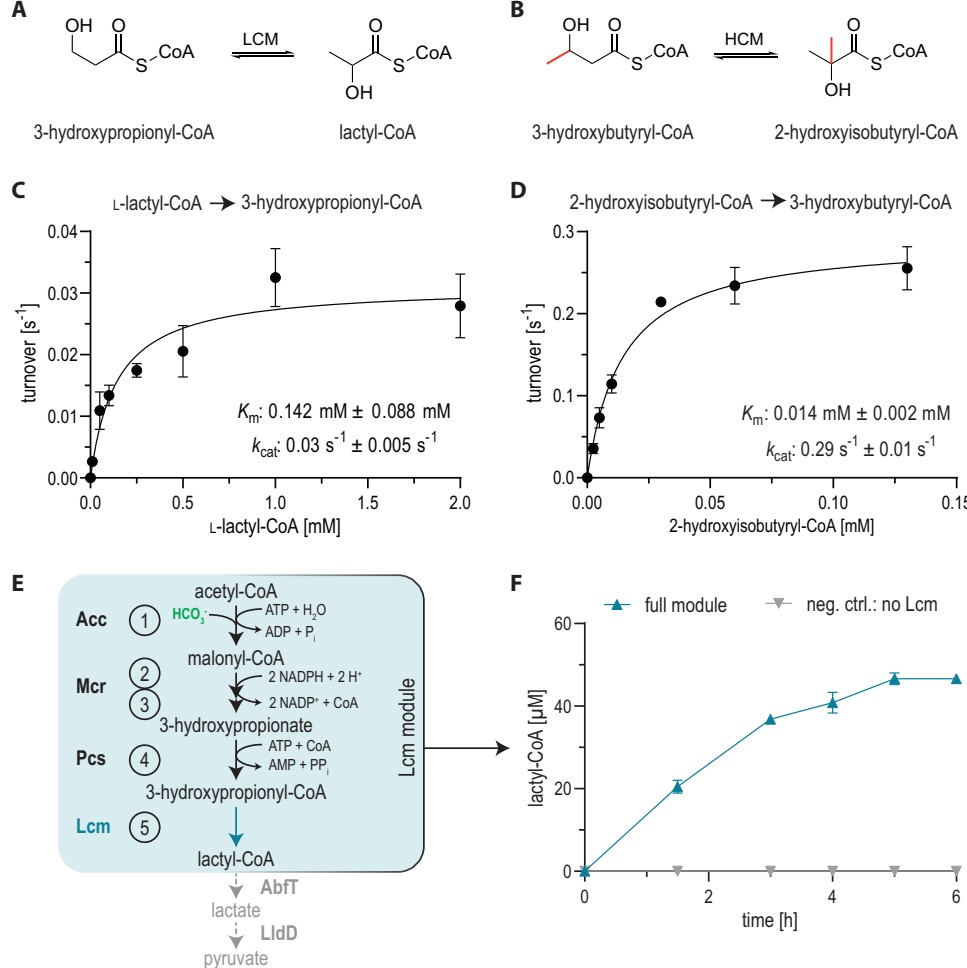

**Fig. 2 | Identification of candidate enzymes to serve as scaffold for Lcm development. A** The promiscuous conversion of 3-hydroxypropionyl-CoA to L-lactyl-CoA (based on B12-dependent substrate radicalization and subsequent carbon backbone rearrangement). **B** Native reaction of the 2-hydroxyisobutyryl-CoA mutases (Hcm) interconverting 3-hydroxybutyryl-CoA and 2-hydroxyisobutyryl-CoA[31]. **C** Promiscuous activity of the wild-type *B. massiliosenegalensis* Hcm with L-lactyl-CoA. Measurements of *n* = 3 are shown with standard deviation. **D** Activity of the Hcm from *B. massiliosenegalensis* with the native substrate 2-hydroxyisobutyryl-CoA. Note that the axes are different from those in panel C. Measurements of *n* = 3 are shown with standard deviation. **E** The Lcm module for the mutase-dependent conversion of acetyl-CoA to lactyl-CoA, which can yield pyruvate. **F** Lactyl-CoA formation from acetyl-CoA via the Lcm module including the ligase domain of propionyl-CoA synthase (Pcs). Measurements of *n* = 3 are shown with standard deviation. Source data are provided as a Source Data file.

## Results

### Design of an oxygen-tolerant, CO₂-assimilating acetyl-CoA assimilation route

To establish a CO₂-assimilating, oxygen-tolerant route from acetyl-CoA to pyruvate, we envisioned using the carboxylation of acetyl-CoA to malonyl-CoA, followed by reduction of the latter to 3-hydroxypropionate (3-HP) and subsequent activation into 3-HP-CoA. We considered the conversion of 3-HP-CoA into lactyl-CoA by a new-to-nature enzymatic activity, and lastly, the conversion of lactyl-CoA into lactate, the precursor of the target metabolite pyruvate (Lcm module, highlighted blue in Fig. 1, shown in detail in Supplementary Fig. 2). Although C4 compounds are not part of the Lcm module, they can easily be synthesized from pyruvate via another carboxylation reaction (by pyruvate carboxylase or PEP carboxylase). The Lcm module outperforms most other acetyl-CoA assimilation routes in terms of thermodynamic favorability (max-min driving force, MDF)[23–25] and ATP requirement (Supplementary Data 1, Supplementary Fig. 2). Furthermore, it is the shortest aerobic route for the synthesis of pyruvate (6 reactions) or oxaloacetate (7-8 reactions) from acetyl-CoA. There are only two shorter alternatives (Supplementary Fig. 2): a natural one involving pyruvate synthase, and a synthetic one, the alanine aminomutase route, which has not yet

been demonstrated experimentally[11]. However, both rely on highly oxygen-sensitive (and partially inefficient) biocatalysts[26–28], which makes their application in biotechnological standard platform organisms highly challenging.

### In vitro screening of enzyme candidates for Lcm activity

To identify suitable candidates for the direct interconversion of 3-HP-CoA and lactyl-CoA (Fig. 2A), we aimed to employ the radical mechanism of O₂-tolerant B12-dependent CoA-thioester mutases. We focused on 2-hydroxyisobutyryl-CoA mutases[29–33] (Hcm) that catalyze the reversible isomerization of 3-hydroxybutyryl-CoA and 2-hydroxyisobutyryl-CoA, which are structurally similar to 3-HP-CoA and lactyl-CoA, respectively (Fig. 2B). We heterologously produced six homologs (Supplementary Fig. 4A), of which we could successfully purify only the three candidates that had already been characterized previously[29,30,34], namely the 2-hydroxyisobutyryl-CoA mutases from *Aquincola tertiaricarbonis*, *Kyrpidia tusciae* and *Bacillus massiliosenegalensis* (also referred to as *Robertmurraya massiliosenegalensis*)[35–37].

Estimation of reaction thermodynamics via eQuilibrator[24] predicted that the isomerization of 3-HP-CoA and lactyl-CoA is fully reversible ($\Delta_r G'° = -4.4 \pm 1.6$ kJ/mol; equilibrium constant K′eq = 6 in the lactyl-CoA forming direction, estimated at physiological pH 7.5 and

ionic strength 0.25 M). Note that 3-HP-CoA isomerization yields lactyl-CoA, which is a chiral product (L- or D-lactyl-CoA). To test for Lcm activity and determine the reaction stereospecificity, we thus first screened our three candidates for 3-HP-CoA forming activity, starting from enantiopure L- or D-lactyl-CoA (Supplementary Fig. 4B). At 30 °C, the enzyme from the mesophilic organism *B. massiliosenegalensis* showed the highest activity with L-lactyl-CoA (Fig. 2C), while the mutase from the thermophilic organism *K. tusciae* had about 2-fold lower activity in these conditions. In contrast, the enzyme candidate from *A. tertiaricarbonis* had no detectable activity with L-lactyl-CoA, despite the fact that we confirmed functionality of this purified enzyme with its native substrate 2-hydroxyisobutyryl-CoA (Supplementary Fig. 4C). None of the mutases showed detectable activity with D-lactyl-CoA.

Due to its mesophilic properties and the basal activity with L-lactyl-CoA, we decided to further test the Hcm from *B. massiliosenegalensis* and confirmed that the enzyme also catalyzed lactyl-CoA formation from 3-HP-CoA (Supplementary Figs. 4D–F) and determined the kinetic parameters for both the native and promiscuous reaction, forming 3-hydroxybutyryl-CoA or 3-HP-CoA, respectively (Fig. 2C, D).

With its native substrate 2-hydroxyisobutyryl-CoA, the *B. massiliosenegalensis* mutase showed a relatively low turnover frequency ($k_{cat} = 0.29 \pm 0.01\,\mathrm{s}^{-1}$) compared to an average enzyme[38]. The enzyme activity with L-lactyl-CoA was even slower ($k_{cat} = 0.03 \pm 0.01\,\mathrm{s}^{-1}$), and the $K_m$ value for L-lactyl-CoA ($0.14 \pm 0.09\,\mathrm{mM}$) was higher than for the native substrate 2-hydroxyisobutyryl-CoA (Fig. 2C, D). Overall, these experiments established L-lactyl-CoA mutase (Lcm), but also indicated that Lcm activity was very low and likely rate-limiting for any application that requires high flux through this reaction in vitro or in vivo.

## The reconstituted Lcm module converts acetyl-CoA into lactyl-CoA in vitro

In order to test whether Lcm would be functional in our new-to-nature acetyl-CoA assimilation pathway, we next sought to reconstitute the enzyme cascade in vitro to test its feasibility. The core sequence of the pathway encompasses five reactions (reactions #1–5, Fig. 2E) converting acetyl-CoA to lactyl-CoA, which we will refer to as the **Lcm module** in the following (Fig. 2E). Besides Lcm from *B. massiliosenegalensis* (reaction #5), we chose propionyl-CoA carboxylase (PccD407I from *Methylobacterium extorquens*[39]) for the carboxylation of acetyl-CoA (reaction #1) and bi-functional malonyl-CoA reductase from *Chloroflexus aurantiacus*[40] for the reduction of malonyl-CoA into 3-HP (#2-3, Fig. 2E). For the activation of 3-HP into 3-HP-CoA (reaction #4, Fig. 2E), we considered two alternative designs, a CoA-transferase route and a CoA ligase route.

For the CoA-transferase route, we tested four different CoA-transferases (AbfT from *Clostridium aminobutyricum* (UniProt ID Q9RM86), Frc from *Oxalobacter formigenes* (UniProt ID O06644), Pct from *Cupriavidus necator* (UniProt ID Q0K874) and *E. coli* YfdE (UniProt ID P76518). Of these, the 4-hydroxybutyrate CoA-transferase AbfT, and the propionyl-CoA-transferase Pct performed best with 3-HP as CoA-acceptor and acetyl-CoA as CoA-donor (Supplementary Fig. 5A). Notably, these enzymes were also capable of recycling the CoA moiety between 3-HP-CoA and lactyl-CoA by using 3-HP-CoA as CoA-donor and L-lactate as CoA-acceptor, albeit at lower catalytic efficiency (Supplementary Fig. 5B).

Next, we evaluated whether the activation of 3-HP by AbfT would also be functional within the context of the complete Lcm module (Fig. 2E). Notably, acetyl-CoA is not only the preferred CoA-donor for 3-HP, but also the substrate of acetyl-CoA carboxylase. Therefore, we first evaluated whether the partial cascade with AbfT would yield sufficient concentrations of the Lcm substrate 3-HP-CoA. However, acetyl-CoA became quickly depleted and was likely not available at sufficient concentrations for 3-HP activation, as indicated by very low 3-HP-CoA

pools (~3 μM) that we observed in our experiments (Supplementary Fig. 5C), indicating a general limitation of the CoA-transferase route.

For the CoA ligase route, we substituted AbfT with a propionyl-CoA ligase (Pcl) from *C. necator* that was previously shown to catalyze the activation of 3-HP[39]. The use of Pcl in the context of the complete Lcm module did not only lead to more than ten times increased 3-HP-CoA levels (~40 μM after 2 h, Supplementary Fig. 5D), but also resulted in the formation of 15 μM lactyl-CoA from 1 mM acetyl-CoA after 2 h, which is equivalent to 1.5% yield (Supplementary Fig. 5D). Despite this improvement, however, we still observed an accumulation of 3-HP, which suggested that the activation of 3-HP to 3-HP-CoA remained a limiting bottleneck. Since Pcl suffers from a high $K_m$ towards 3-HP[39], we decided to replace it with the ligase domain of the propionyl-CoA synthase from *Erythrobacter sp.* NAP1 (Pcs, Fig. 2E) that catalyzes the same reaction[41]. The use of Pcs increased the 3-HP-CoA concentration to 100 μM after 2 h, but did not further increase lactyl-CoA levels. Lactyl-CoA levels reached ~20 μM after 2 h, which let us assume that Lcm activity was the bottleneck of the Lcm module (Supplementary Fig. 5E). This was supported by the fact that while 3-HP-CoA concentration continued to increase up to 500 μM within 22.5 h, the Lcm module with Pcs produced only about 50 μM L-lactyl-CoA (5% conversion) and peaked at about 6 h (Fig. 2F, Supplementary Fig. 5E).

Having demonstrated the feasibility of the Lcm module and confirmed Lcm as the limiting bottleneck in the cascade, we next aimed to engineer improved Lcm variants. However, all our rational efforts that targeted different first-shell active site residues, which we expected to coordinate the substrate (Supplementary Fig. 6A), rendered the protein inactive (Supplementary Fig. 6B), so we turned our attention to a random mutagenesis-based approach.

## Design of a growth-coupled selection strain to test for Lcm activity in vivo

To improve Lcm activity, we sought to generate different Lcm variants by random mutagenesis and screen them with a selection strain that would rely on the mutase activity for growth. Because of the low catalytic activity of Lcm (~$0.03\,\mathrm{s}^{-1}$), we aimed to design a strain with relatively low selective demand. For this goal, we repurposed a β-alanine auxotrophy[42,43].

In *E. coli*, low amounts of the non-proteinogenic amino acid β-alanine are essential for CoA biosynthesis. Natively, *E. coli* produces the required β-alanine via decarboxylation of L-aspartate[44]. Deletion of the gene encoding the corresponding enzyme (*panD*) creates an auxotrophy, which can be rescued by supplementation of β-alanine[43]. We aimed to replace the native *panD*-based route for production of β-alanine by an Lcm-dependent pathway proceeding from L-Lactate via 3-HP (Fig. 3A). Initially, we confirmed that 3-HP can be converted into β-alanine through the chromosomal expression of two heterologous helper genes, encoding a 3-HP dehydrogenase and β-alanine transaminase. As anticipated, growth of the β-alanine auxotrophic strain could be rescued by addition of 3-HP, with the lower limit of detection reaching a low micro-molar range, confirming the general feasibility of our screening strategy (Supplementary Fig. 7A, B).

We next sought to introduce Lcm in combination with a CoA-transferase and test for growth on L-lactate (Fig. 3A). We expressed Lcm together with its native chaperone MeaB using a previously reported expression system with a medium-copy pBBR1 origin[30] and tested two different CoA-transferases, AbfT and Pct, from our in vitro assays (Supplementary Fig. 7C). Overexpression of the genes encoding AbfT and Lcm allowed growth of the selection strain with L-lactate instead of 3-HP (Fig. 3B). This growth was only observed upon expression of both genes and was strictly dependent on the addition of both L-lactate and $B_{12}$ (supplemented as cyanocobalamin to the medium), providing evidence for Lcm-activity in vivo (Supplementary Fig. 7C, D). We tested three different AbfT expression levels (chromosomal expression with two different ribosomal binding sites (RBS

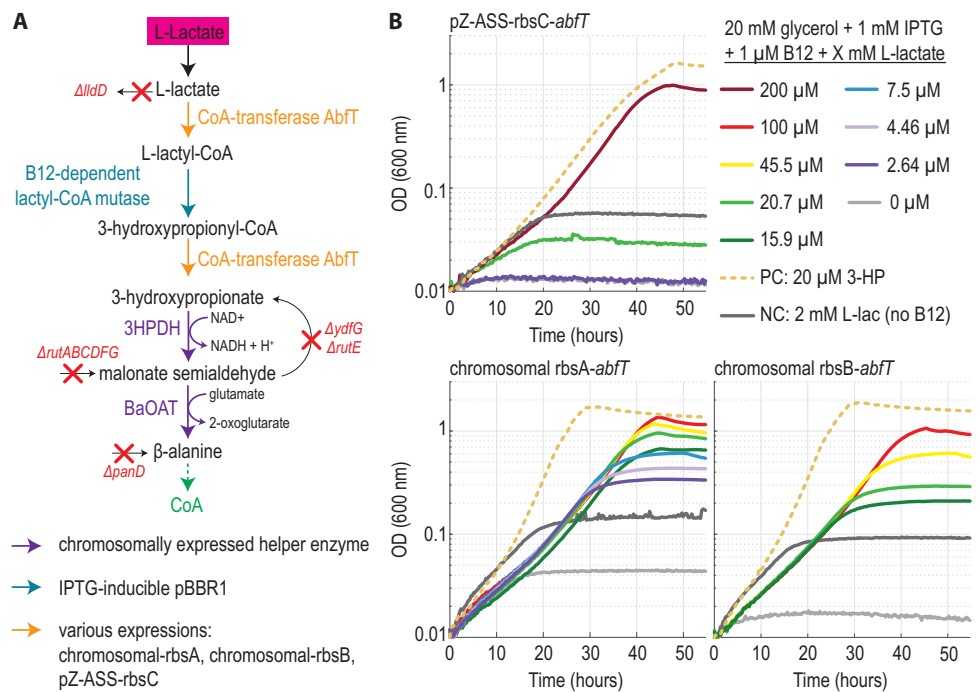

**Fig. 3 | Basal Lcm activity suffices to rescue a β-alanine auxotroph selection strain in vivo. A** A strain deleted in native β-alanine biosynthesis routes and L-lactate dehydrogenase (β-alanine auxotroph, see Supplementary Data 2 for details) relies on joint activity of a CoA-transferase and Lcm to convert L-lactate to 3-hydroxypropionate (3-HP). The helper enzymes 3-HP dehydrogenase (3-HPDH) and β-alanine transaminase (BaOAT) convert 3-HP to β-alanine, which is required for CoA biosynthesis. **B** Expression of the CoA-transferase AbfT (expression

indicated in each graph) and the Lcm (pBBR1-Lcm) reconstitute the ability of the β-alanine auxotroph to grow with L-lactate (colored lines) instead of 3-HP (dotted lines). No growth was observed in negative controls lacking L-lactate (light gray) or B12 (dark gray), respectively. Chromosomal expression of AbfT with ribosomal binding site (rbs) "A" results in the highest biomass yields with the lowest amounts of L-lactate. Source data are provided as a Source Data file.

"A", higher expression level and RBS "B", lower expression level)[45], or expression from a low-copy plasmid). We found that albeit all strains could grow in a lactate-dependent manner, AbfT expression from the genome with RBS "A" resulted in the highest $OD_{600}$ values at low L-lactate concentrations (Fig. 3B). Notably, with all constructs, a few doublings were observed in medium lacking $B_{12}$, before growth ceased, which can be attributed to utilization of residual intracellular $B_{12}$ from the preculture[46].

To independently verify operation of the Lcm pathway, we used $^{13}$C-labeling with $^{13}C_3$-L-lactate (Supplementary Fig. 8) and analyzed the CoA isotope pattern with high resolution-mass spectrometry. While the labeling pattern of CoA was relatively complex, almost no CoA appeared unlabeled, single- or double-labeled in the β-alanine auxotroph with the Lcm module (Supplementary Fig. 8B). This was in contrast to the wild type and in line with incorporation of the triple-labeled unit from $^{13}C_3$-L-lactate. Furthermore, detailed analysis of triple-labeled CoA showed that in the β-alanine auxotroph with the Lcm module, the label indeed resided in the β-alanine-containing fragment (Supplementary Fig. 8C). In summary, both growth behavior and labeling distribution observed for the β-alanine auxotroph confirmed Lcm-dependent growth and thus Lcm activity in a metabolic context in vivo.

**Combining in vivo hypermutation with growth-coupled selection identifies improved Lcm variants**
Next, we created a library of Lcm variants by randomized mutagenesis targeting the Lcm expression plasmid. To combine library generation and growth-coupled selection for optimized Lcm variants, we performed targeted in vivo hypermutation via the eMutaT7 system[47] (Supplementary Fig. 9A) followed by subsequent short-term adaptive evolution in the β-alanine auxotroph (Supplementary Fig. 9B). Unfortunately, we did not obtain growth of the β-alanine auxotroph

expressing Lcm from the eMutaT7 target plasmid (pMutaT7-Lcm, see Methods[48]) in selective conditions when the mutator plasmid (pMutator; see Methods[47]) was co-expressed. Therefore, we produced two independent libraries (#1 and #2) of Lcm variants by induction of the eMutaT7 system in non-selective conditions (LB medium) overnight. To enrich these diversified populations for the most active Lcm variants, we subsequently propagated these libraries in medium selecting for Lcm activity and cured them from the pMutator plasmid (Supplementary Fig. 9B).

When no further growth improvement was observed during the selective propagations, growth of both evolved populations was first compared to that of the parental strain and the pBBR1-Lcm containing strain in a plate reader (Supplementary Fig. 10). Single clones were isolated and tested on selective medium, where the mean growth rate of all growing clones isolated from both libraries was higher than that of the non-evolved parental strain (expressing Lcm from the single-copy MutaT7 target plasmid) as well as the initially characterized strain expressing Lcm from the medium-copy pBBR1-plasmid (Fig. 4A). Sequencing of the Lcm expression plasmids from these strains showed four distinct variants which all carried mutations in the large Lcm-subunit (Lcm-A) and/or in the chaperone MeaB (Table 1).

No mutations were found in the promoter region of the plasmids, indicating that changes in the Lcm sequence were responsible for growth improvement rather than merely changes in expression level. To verify that these mutations were responsible and sufficient for the observed phenotype, we re-introduced the mutated plasmids into the non-evolved parental background (β-alanine auxotroph with chromosomal rbsA-abfT expression). In each case, this reproduced the observed growth phenotype (Fig. 4B and Supplementary Fig. 11), confirming that no additional adaptation was required.

Mapping the mutations onto the structure of a homolog, the *A. tertiaricarbonis* Hcm (PDB: 4r3u, RMSD 0.871 Å[31]), showed that no

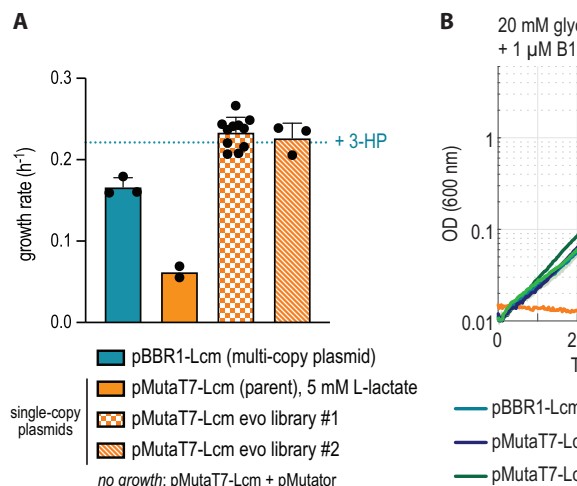

**Fig. 4 | In vivo hypermutation and growth-coupled selection resulted in improved Lcm-dependent growth. A** Growth rate of all Lcm-dependent strains with L-lactate. The evolved isolates have a significantly improved growth rate compared to previous strains. The blue dotted line indicates the growth rate observed for the pBBR1-Lcm containing strain with 3-HP (positive control medium), which we defined as the upper threshold for expected growth rate improvement. For both libraries, the mean growth rate of all isolated and growing single clones is shown with standard deviation ($n = 11$ for library 1, $n = 3$ for library 2). Number of replicates for pBBR1-LCM = 3, for pMutaT7 = 2. Since the pMutaT7-Lcm (parent) strain failed to grow with 1 mM L-lactate, the growth rate with 5 mM L-lactate is shown (solid orange bar), whereas all other strains were grown with 1 mM L-lactate. **B** Growth of β-alanine auxotrophic strains expressing the previously used multi-copy pBBR1 plasmid or pMutaT7-Lcm variants isolated from mutants with improved growth rates. Note that the mutant plasmids were reintroduced to the unevolved genetic background. Note that the pMutaT7-Lcm (parent) strain (orange curve) failed to growth with 1 mM L-Lactate and only grew with 5 mM L-lactate (Fig. 4A, Supplementary Fig. 10). Source data are provided as a Source Data file.

amino acid change directly pointed at the substrate or towards the active site, indicating rather subtle changes to the active site geometry in these three Lcm variants (Supplementary Fig. 12). Additionally, none of the evolved mutations appear to be located at the binding interface of Lcm and its chaperone MeaB, as modeled using AlphaFold2 (Supplementary Fig. 13).

### Lcm variants show improved activity in vitro

To test the effect of the different mutations on catalytic activity, we purified the individual Lcm variants and determined their kinetics. In all three Lcm variants, both $k_{cat}$ and $K_m$ were improved (Fig. 5A, B), resulting in an increase of catalytic efficiency ($k_{cat}/K_m$) by 5-fold (variant P163L and variant E102K) and 10-fold (variant V92I E137K) compared to the wild type (Fig. 5B, Table 2), when testing in the direction of 3-HP-CoA formation.

This trend was confirmed also in the reverse direction, i.e. when measuring lactyl-CoA formation from 3-HP over time (Fig. 5C). In these assays, however, we noticed that P163L appeared to be more instable and was repeatedly inactive after prolonged incubation on ice.

Having identified three improved Lcm variants, we finally tested these enzymes in the context of the complete Lcm module in vitro (Fig. 5D). Notably, in these assays, the variant E102K performed better than the wild type, while the variants V92I E137K and P163L compared similar or even worse (P163L), indicating that not only kinetics, but also enzyme stability (i.e. P163L) might be an important factor in vitro.

Given the sensitivity of $B_{12}$-dependent enzymes to inactivation by molecular oxygen, particularly in vitro, we hoped that supplying more $B_{12}$ could improve Lcm activity and thus lactyl-CoA yield of the Lcm module in our in vitro assay. To test this hypothesis, we first compared the mutase turnover with 12.5 µM $B_{12}$ (2.5x excess over enzyme) and 500 µM (100x excess), and found that higher $B_{12}$ concentrations increased enzyme turnover by 30% (Supplementary Fig. 14A). However, when we tested the entire cascade with 12.5 µM (2.5x excess over enzyme), 100 µM (20x excess) and 500 µM $B_{12}$ (100x excess), the $B_{12}$-dependent increase in lactyl-CoA concentrations after six hours was much less pronounced (Supplementary Fig. 14B). Since high $B_{12}$-concentrations were previously shown to impair in vitro operation of the CETCH and THETA cycles[9,49], we hypothesized that one of the other Lcm module enzymes might be inhibited by the additional $B_{12}$. Indeed, we saw that Pcs-dependent 3-HP-CoA formation from 3-HP decreased with increasing $B_{12}$ concentrations (Supplementary Fig. 14B). To test whether the Pcs would suffer from $B_{12}$-dependent inhibition, we tested only Pcs and wild-type Lcm on 1 mM 3-HP and found a severe decrease in 3-HP-CoA formation that was correlated to the $B_{12}$ concentration (Supplementary Fig. 14C). In summary, these findings indicate that $B_{12}$ stability and complex interactions of the cofactor with the reaction cascade is a major drawback in vitro that would need to be further engineered in the future.

## Discussion

In this work, we established the Lcm module, a new-to-nature metabolic pathway for the $CO_2$-assimilating, oxygen-tolerant conversion of acetyl-CoA to pyruvate, which relies on a new-to-nature enzymatic activity as the key reaction: the $B_{12}$-dependent conversion of lactyl-CoA into 3-hydroxypropionyl-CoA by lactyl-CoA mutase (Lcm).

Of several potential Lcm candidates, we identified 2-hydroxyisobutyryl-CoA mutase (Hcm) from *B. massiliosenegalensis* as a functional scaffold with promiscuous Lcm side-activity in vitro and in vivo. In the context of the complete Lcm module, this basal activity sufficed to achieve a 5% conversion of acetyl-CoA into lactyl-CoA within six hours in vitro, but appeared to be a kinetic bottleneck for the pathway. Using a combination of growth-coupled selection and

### Table 1 | Mutations obtained by hypermutation and short-term adaptive evolution

| Library | Plasmid ID | Mutations on pMutaT7-Lcm | | |
| --- | --- | --- | --- | --- |
| | | MeaB | Lcm-A | Lcm-B |
| #1 | pMutaT7-Lcm evo 1 | P52L | P163L | - |
| #2 | pMutaT7-Lcm evo 2 | - | V92I E137K | - |
| | pMutaT7-Lcm evo 3 | silent G135G (tgg → ggg) | - | - |
| | pMutaT7-Lcm evo 4 | P195S | E102K | - |

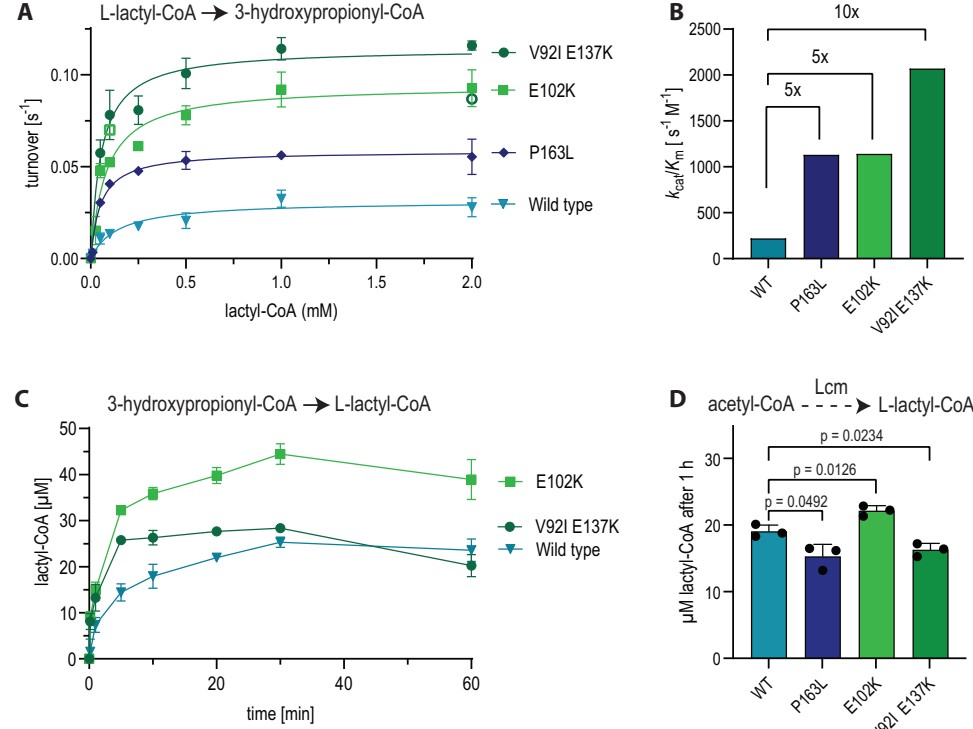

**Fig. 5 | Activity of Lcm variants in the 3-hydroxypropionyl-CoA forming direction.** All assays were performed using equimolar amounts of large A subunit (the respective variant) and wild-type small B subunit (5 μM of each were used). **A** Michaelis Menten Kinetics of the wild-type Lcm and variants P163L, V92I E137K, E102K with ʟ-lactyl-CoA. Points denote mean values of technical triplicates with standard deviation. Outliers (defined as values that differed by > 15% from the mean; shown as empty symbols) were excluded for curve-fitting. **B** Catalytic efficiency of all mutant variants compared to the wild-type Lcm. **C** Activity of all Lcm variants in the lactyl-CoA forming direction. Variant P163L was inactive and is therefore not shown, it was purified again for further testing. Measurements of n = 3 are shown with standard deviation. **D** Lcm module with each Lcm variant. The reaction rate was determined by quantifying lactyl-CoA formation from acetyl-CoA after one hour. Pcs was used for 3-HP-CoA synthesis, and 100 μM B₁₂ were added to the assay. An unpaired two-sided t-test with Welch's correction was used to analyze statistical significance, p-values for each comparison are given in the graph. Measurements of n = 3 are shown with standard deviation. Source data are provided as a Source Data file.

hypermutation by the eMutaT7 system[47,48,50,51], we improved the catalytic efficiency ($k_{cat}/K_m$) of Lcm ten-fold.

The Lcm activity expands the existing metabolic solution space[52], and can be flexibly applied in several synthetic biology contexts. Firstly, Lcm provides an alternative metabolic route for the microbial production of the bioplastic precursor 3-hydroxypropionate (3-HP)[53,54] or downstream bioproducts such as β-alanine and malonate. Previous approaches to the biosynthesis of these compounds either (i) depended on glycerol as a feedstock via an oxygensensitive pathway involving glycerol dehydratase[55], (ii) relied on decarboxylation of aspartate[56,57], or (iii) required ATP-dependent carboxylation of acetyl-CoA[58]. In contrast to these routes, the Lcm module could allow biosynthesis of 3-HP directly from pyruvate via lactate, which can be produced at high yields via canonical sugar fermentation by industrially well-established hosts such as *E. coli*. Indeed, the potential of 3-HP production from lactate was recognized in previous theoretical studies[53,54], however, the Lcm module now provides an experimental proof-of-principle for this production route.

Secondly (and as initially intended), the Lcm module can serve as an extension of aerobic C1-assimilating pathways that yield acetyl-CoA as their direct product, such as the THETA cycle or the serine-threonine cycle[9,12], which allows to improve their biomass yields through co-fixation of CO₂ during downstream acetyl-CoA assimilation (Supplementary Data 1). Indeed, carbon-conserving routes to convert intermediates of CO₂ fixation to central metabolites have been explored before, for example by engineering a non-oxidative glycolysis or the reverse glyoxylate shunt[59,60].

Thirdly, the Lcm module could assist in the microbial production of value-added chemicals from compounds feeding directly into acetyl-CoA, such as ethanol and acetate[61,62]. These two-carbon feedstocks are already sustainably produced at industrial scale via syngas fermentation[63], and can also be derived from waste streams, degradation of lignocellulosic biomass[64] or via abiotic, electrochemical synthesis[65,66].

Lastly, the Lcm module can be used as a core sequence of synthetic CO₂-fixation pathways: the Lcm-dependent conversion of 3-HP-CoA to lactyl-CoA could allow implementing a variant of the previously described malonyl-CoA-oxaloacetate-glyoxylate cycles (Supplementary Fig. 3)[11].

However, for the majority of these applications, the Lcm activity must be further improved. Both acetyl-CoA assimilation and synthetic CO₂ fixation cycles require high flux. Despite the significant performance improvement compared to the wild-type Lcm, the turnover of the best Lcm variant described here ($k_{cat} = 0.11 \pm 0.009\ \text{s}^{-1}$) is still two orders of magnitude below that of an average enzyme ($k_{cat} \approx 10\ \text{s}^{-1}$)[38].

**Table 2 | Kinetic parameters of all Lcm variants in the 3-HP-CoA forming direction**

| Variant | $k_{cat}$ (s⁻¹) | $K_m$ (μM) | $k_{cat}/K_m$ (s⁻¹ M⁻¹) |
|---|---|---|---|
| V92I E137K | 0.11 ± 0.009 | 55 ± 22 | 2.07 × 10³ |
| E102K | 0.09 ± 0.009 | 83 ± 36 | 1.14 × 10³ |
| P163L | 0.06 ± 0.003 | 52 ± 15 | 1.13 × 10³ |
| Wild type | 0.03 ± 0.005 | 142 ± 88 | 0.22 × 10³ |

The values reflect mean ± standard deviation for n = 3.

Thus, all future applications would benefit from Lcm variants with higher turnover than achieved so far. We envision that such improvements could be achieved by additional rounds of mutagenesis and subsequent screening using the workflows and selections established here, or in selections with higher flux demand. Since the Hcm scaffold we chose is slow even with its native substrate, we expect that sampling further candidates from the natural diversity of Hcm homologs may aid the identification of faster Lcm catalysts[29,32,33,67]. Indeed, related $B_{12}$-dependent mutases achieve much higher turnover frequencies. For instance, methylmalonyl-CoA mutase (Mcm) is part of central metabolism in multiple organisms and has been reported to achieve $k_{cat}$ values above $200\,s^{-1}$ (e.g. *Methylobacterium extorquens* Mcm with a $k_{cat}$ of $255\,s^{-1}$)[68,69]. Therefore, we reason that the reaction mechanism itself should allow evolving the lactyl-CoA mutase towards even higher turnovers than those observed for the native reaction of Hcm[29,32,33].

For future applications in a cell-free context, optimization of the stability and oxygen-tolerance of Lcm may represent another important target, since inactivation of $B_{12}$-dependent mutases by molecular oxygen is well-described, especially in vitro[69–74]. We emphasize that applications of Lcm in vivo allow the co-expression of chaperones (MeaB)[69,75], salvage of damaged coenzyme $B_{12}$[76] and improved protection of the coenzyme from photolysis and oxidative damage in the reductive intracellular environment. Applications in vivo thus likely present a more promising environment for future implementations of Lcm compared to in vitro systems.

In summary, Lcm and the Lcm module open possibilities for metabolic engineering and might – upon further improvement in the future – become valuable additions to ongoing efforts of creating new C1-assimilating metabolic pathways for a sustainable bioeconomy.

## Methods

### MDF analysis
To determine the net reaction equations of all acetyl-CoA assimilation modules and evaluate their thermodynamic feasibility in physiological conditions, Max-Min Driving Force (MDF) analysis was conducted using the Python packages equilibrator API and equilibrator pathway (version 0.6.0 for both)[25]. Changes in Gibbs Free Energy were estimated using the Component Contribution Method[25]. For all carboxylation reactions, $CO_2$ (rather than bicarbonate) was considered as substrate since its concentration is pH-independent[9]. Default values were used, i.e. the pH was set to 7.5, magnesium concentration was set to 1 mM (pMg = 3), ionic strength 0.25 M, and metabolite and cofactor concentrations were constrained to physiologically relevant concentrations (1 μM – 10 mM)[77].

### Protein production and purification
To produce recombinant proteins, *E. coli* BL21 (DE3) (for all proteins except the mutases) or *E. coli* ArcticExpress (DE3) (all mutase variants and subunits) were transformed via heat-shock with the respective plasmids and plated on LB plates with the respective antibiotics. The cells were scraped off the plate and inoculated in TB (12 g/L tryptone, 24 g/L yeast extract, 17 mM $KH_2PO_4$, 72 mM $K_2HPO_4$, 0.5% (v/v) glycerol) with antibiotic. For purification from ArcticExpress (DE3), gentamycin was supplied as well. The cells were grown at 37 °C, 120 rpm until an $OD_{600}$ of -0.7-0.9 was reached. The cultures were cooled down to 25 °C (BL21(DE3) or 16 °C (ArcticExpress (DE3)) and induced with 500 μM IPTG (pET) or 50 μg/L anhydrotetracycline (pASG-IBA43). The culture was incubated overnight and harvested the next day (4000 × *g*, 10 min, 10 °C). The pellet was resuspended in Buffer A (50 mM HEPES pH 7.5, 500 mM KCl) and sonicated on ice to lyse the cells (KE76 probe, 1 min, 01 s on 01 s off, 3x in total) or microfluidized 3 times with 18,000 psi. The lysed cells were centrifuged (100,000 × *g*, 45 min, 4 °C) to separate soluble and insoluble fraction. The soluble fraction was loaded onto Ni-NTA agar resin (Macherey-Nagel) that was previously equilibrated with wash buffer (50 mM HEPES pH 7.5, 75 mM Imidazole, 500 mM KCl). The resin was washed with 70 mL wash buffer and then eluted with 2 mL Buffer B (50 mM HEPES pH 7.5, 500 mM Imidazol, 500 mM KCl). The proteins were desalted into Buffer D (50 mM HEPES/KOH pH 7.5, 150 mM KCl) using an ÄKTA start system with HiTrap Desalting columns (GE healthcare) or PD-10 columns (Cytiva) and concentrated on Amicon filters. The proteins were flash frozen in liquid $N_2$ and stored at −70 °C. PCC D407I, a PCC variant described by Scheffen et al. [78], was purified by His-tag purification as described above.

### In vitro characterization of Lcm
To measure Lcm-based formation of 3-HP-CoA from ʟ-lactyl-CoA, the Lcm reaction was coupled to PhaJ-mediated dehydration of 3-HP-CoA to acrylyl-CoA, which was reduced to propionyl-CoA in an NADPH consuming reaction catalyzed by Etr1p from *Saccharomyces cerevisiae*[41]. For detection of Hcm based formation of (*R*)−3-hydroxybutyryl-CoA from 2-hydroxyisobutyryl-CoA, the same assay was used with PhaJ producing crotonyl-CoA from (*R*)−3-hydroxybutyryl-CoA and Etr1p producing butyryl-CoA from the latter in an NADPH reducing manner. Both PhaJ and Etr1p had previously been shown not to be rate limiting. 12.5 μM coenzyme $B_{12}$ were supplied in all assays, which additionally contained 50 mM NaPO4 pH 7 and 5 μM of each mutase subunit. B12 and both mutase subunits were mixed and incubated in the assay cuvette for one minute before activity measurements. The NADPH consumption was followed by measuring the change of absorption at 340 nm in a Cary60 UV-Vis spectrophotometer (Agilent, Santa Clara, CA, USA) with the kinetics program with 0.5 s save time. A reaction volume of 100 μL was used in a High Precision Cell 10 mm Light Path quartz cuvette (Hellma Analytics, Müllheim, Germany) at 30 °C. Initial velocity measurements were performed in triplicates, and turnover frequencies were calculated based on Lambert-Beer's law with $\varepsilon_{(340\,nm)} = 6.22\,mM^{-1}\,cm^{-1}$ and fit using the Michaelis-Menten function in Prism 8 (Graphpad, San Diego, California, USA). To test formation of lactyl-CoA, 50 mM NaPO4 pH 7, 10 mM $MgCl_2$, 10 μM of both subunits, 15 μM coenzyme $B_{12}$ and 1.5 mM 3-HP-CoA were assembled. The reaction was run at 30 °C and quenched with formic acid to a total concentration of 10% (v/v) and subsequently analyzed with LC-MS.

### In vitro screening of CoA-transferases
Screening of candidates that can transfer the CoA moiety from 3-HP-CoA to ʟ-lactyl-CoA was performed using LC-MS assays. Here, 1 mM of neutralized acid (ʟ-lactic acid or 3-hydroxypropionic acid) was incubated in 100 mM HEPES pH 7.5, 10 mM $MgCl_2$ and 100 μM of CoA donor (acetyl-CoA, ʟ-lactyl-CoA or 3-HP-CoA). The reaction was started by the addition of 1 μM of the respective CoA-transferase. The assay was run at 30 °C for 1 h. Samples were quenched in formic acid to a final concentration of 10% (v/v) and analyzed via LC-MS.

### In vitro testing of the acetyl-CoA assimilation module
In order to assess the capability of the mutase to be part of an acetyl-CoA assimilation pathway, we assembled the pathway in vitro. In brief, 100 mM HEPES pH 7.5, 0.5 mM TECEP, 5 mM $NaHCO_3^-$, 10 mM $MgCl_2$, 1 μM PCC D407I, 1 μM MCR, 5 μM BmHcmA (and variants), 5 μM BmHcmB, 12.5 μM coenzyme $B_{12}$, 2.5 mM NADPH, 5 mM Glucose-6-P, 1 U G6PDH (Sigma G5885), 2 mM ATP, 5 mM Creatine-P, 1U Creatine Kinase (Sigma C3775) and 1 μM Pcl or 1 μM Pcs were assembled into a tube. For the Pcs reaction, 1 μM adenylate kinase from *E. coli* was added as well. The reaction was run at 30 °C for up to -24 h and started by the addition of acetyl-CoA. Samples were quenched in formic acid to a final concentration of 10% (v/v) and analyzed via LC-MS. If needed, samples were diluted further.

### In vitro testing of $B_{12}$-dependent Pcs inhibition

To determine the effect of $B_{12}$ on Pcs and Lcm activity, we tested their joint activity from 3-HP with varying $B_{12}$ concentrations in vitro. For this, 100 mM HEPES pH 7.5, 0.5 mM TECEP, 10 mM $MgCl_2$, 5 µM BmHcmA (wild-type variant), 5 µM BmHcmB, 2 mM ATP, 5 mM Creatine-P, 1 U Creatine Kinase (Sigma C3775), 2 mM free CoA, 1 µM adenylate kinase from *E. coli* and 1 µM Pcs and coenzyme $B_{12}$ (tested concentrations were 12.5 µM, 100 µM, 500 µM) were assembled into a tube. The reaction was run at 30 °C for up to ~6 h and started by the addition of 1 mM 3-HP. Samples were quenched in formic acid to a final concentration of 10% (v/v) and analyzed via LC-MS. If needed, samples were diluted further.

### CoA thioester synthesis

CoA thioesters were synthesized using the CDI route[79]. Divergent from the published method, the synthesis of lactyl-CoA was stirred overnight on ice after addition of free CoA to CDI-activated lactic acid. Enantiopure acids were used for the synthesis of L- or D-lactyl-CoA. The synthesis product was purified using a preparative Agilent 1260 Infinity HPLC with a Gemini 10 µm NX-C18 110 Å column. The purified CoA thioesters were flash frozen in liquid $N_2$, lyophilized and stored dry at −20 °C. The product identity was confirmed via LC-MS.

### Quantification of lactyl-CoA, 3-hydroxypropionyl-CoA, malonyl-CoA and acetyl-CoA

Quantitative metabolite determination was performed using a LC-MS/MS. Chromatographic separation was performed on an Agilent Infinity II 1290 HPLC system with a Kinetex EVO C18 column (150 × 2.1 mm, 3 µm particle size, 100 Å pore size, Phenomenex) connected to a guard column of similar specificity (20 × 2.1 mm, 3 µm particle size, Phenomenex) at a constant flow rate of 0.2 mL/min and an injection volume of 1 µL. The mobile phase A was 50 mM Ammonium Acetate in water at a pH of 8.1 and phase B was 100% methanol (Honeywell, Morristown, New Jersey, USA) at 40 °C column temperature. The mobile phase profile consisted of the following steps and linear gradients: 0–7 min constant at 5% B; 7–8 min from 5 to 95% B; 8–9 min constant at 95% B; 9–9.1 min from 95 to 5% B; 9.1 to 15 min constant at 5% B. An Agilent 6495 ion funnel mass spectrometer was used in positive mode with an electrospray ionization source and the following conditions: ESI spray voltage 1000 V, nozzle voltage 1000 V, sheath gas 350 °C at 12 l/min, nebulizer pressure 20 psig and drying gas 100 °C at 11 l/min. Compounds were identified based on their mass transition and retention time compared to standards. Chromatograms were integrated using the MassHunter software (Agilent, Santa Clara, CA, USA). Relative abundance was determined based on the peak area. Absolute concentrations were determined based on an external Standard curve. Mass transitions, collision energies, Cell accelerator voltages and Dwell times have been optimized using chemically pure standards. Parameter settings of all targets are given in Supplementary Table 2.

### Quantification of 3-hydroxypropionate and lactate

Quantification of 3-HP and lactate was performed as described above for CoA esters, but at a constant flow rate of 0.1 mL/min with mobile phase A being 0.1% formic acid in water and phase B being 0.1% formic acid in methanol (Honeywell, Morristown, New Jersey, USA) at 25 °C. The injection volume was 1 µL. The profile of the mobile phase consisted of the following steps and linear gradients: 0–6 min constant at 0% B; 6–7 min from 0% to 100% B; 7–8 min constant at 100% B; 8–8.1 min from 100% to 0% B; 8.1–12 min constant at 0% B. An Agilent 6495 ion funnel mass spectrometer was used in negative mode with an electrospray ionization source and the following conditions: ESI spray voltage 2000 V, nozzle voltage 500 V, sheath gas 260 °C at 10 L/min, nebulizer pressure 35 psig and drying gas 100 °C at 13 L/min. Parameter settings of all targets are given in Supplementary Table 3.

### Strain construction

All *E. coli* strains used in this study are listed in Supplementary Data 2. We used strain SIJ488 as a wild-type starting strain, which carries inducible recombinase and flippase genes[80]. Gene deletions were done by either λ-red recombineering or P1-transduction, as described below.

### Gene deletion and genome integration by recombineering

To delete genes by λ-red recombineering, chloramphenicol resistance cassettes with overhangs homologous to the target locus were generated by PCR using KO primers (Supplementary Data 3), the chloramphenicol (Cap) cassette from pKD3 (pKD3 was a gift from Barry L. Wanner (Addgene plasmid #45604; http://n2t.net/addgene:45604; RRID:Addgene_45604)) as template and PrimeStar Max polymerase (Takara Bio, Saint-Germain-en-Laye, France). For gene deletion, the target strains were inoculated in LB and grown to OD ~ 0.4–0.5, followed by addition of 15 mM L-arabinose to induce recombinase gene expression during 45 min cultivation at 37 °C. Then, cells were harvested and washed three times with ice cold 10% glycerol (11,000 rpm, 30 sec, 4 °C). Electroporation was performed using ~300 ng of Cap resistance cassette and 100 µL washed cells (1 mm cuvette, 1.8 kV, 25 µF, 200 Ω). To select for successful gene deletion, cells were plated on LB chloramphenicol. Further deletion confirmation was done by PCR, and antibiotic resistance removal was performed by growing cells to OD ~ 0.3, inducing flippase expression by addition of 50 mM L-rhamnose, cultivation overnight at 30 °C and subsequent isolation of single colonies on LB plates. Successful marker removal was confirmed by testing for antibiotic sensitivity and by PCR on the respective locus. To chromosomally integrate *abfT*, the gene was initially cloned from pZ-ASS into pKD3 (see section "Plasmid construction" below) and subsequently amplified with PrimeStar Max polymerase and primers carrying 50 bp overhangs to the *lldD* locus. To remove residual PCR template, the PCR product was digested with *DpnI* (FastDigest, Thermo Scientific) before purification. The purified PCR product was introduced into the genome via recombineering and successful integration was selected for via antibiotics as described above for deletions.

To chromosomally integrate the 3-HPDH and BaOAT, a cassette with 500 bp overhangs from both sides of the targeted gene was generated. Followed by amplifying upstream overhangs, downstream overhangs and target genes, an overlap PCR was performed. For the integration of BaOAT in SS2, the left and right integration overhang were amplified from pDM4-SS2 using primers SS2-KI_fwd + KI_B or SS2_KI_rvs + KI_E, respectively. BaOAT was amplified from pZ-ASS using the primers KI_C + KI_D. To integrate 3-HPDH in SS9, the left and right integration overhang were amplified from pDM4-SS9 using primers SS9-KI_fwd + KI_B or SS9_KI_rvs + KI_E, respectively. 3-HPDH was amplified from pZ-ASS using the primers KI_C + KI_D. The product of the overlap extension PCR was directly used as Knock-In cassette to perform the genomic integration into the relevant loci (SS2 for BaOAT, SS9 for 3-HPDH)[81] by electroporation as described for deletions by λ-red recombineering.

### Plasmid construction

For gene overexpression, *abfT* (CoA-transferase from *Clostridium aminobutyricum*, UniProt ID Q9RM86) and 3-HPDH (3-hydroxyisobutyrate dehydrogenase *pmHPD* from *Pseudomonas putida*, UniProt ID Q88E02) were codon optimized for *E. coli*'s codon usage[82], and restriction sites relevant for cloning were removed[45] (for codon-optimized sequences, see Supplementary Data 4). A His-tag with *NsiI* restriction site was added to the start of the gene, an *XbaI* restriction site was added behind the stop codon. The genes were synthesized by Twist Bioscience (San Francisco, CA, USA). Cloning was performed in *E. coli* DH5α. All genes were cut via *NsiI* and *XbaI* (FastDigest, Thermo Scientific) and ligated with a pZ-ASS-mCherry[83] vector cut *NsiI* and *NheI* (FastDigest, Thermo Scientific) (thus removing mCherry and

linearizing the backbone) downstream of ribosome binding site "C" (AAGTTAAGAGGCAAGA)[45]. pZ-ASS-*lkBOT* (BaOAT) was obtained from a previous study[84]. For genome integration, the genes were transferred from pZ-ASS to pKD3 by restriction of both vectors with *NsiI* and *BspOI* (FastDigest, Thermo Scientific), gel purification of fragments of the desired size and subsequent ligation of pKD3 backbone with the respective insert. Correct insert sizes were confirmed using DreamTaq polymerase (Thermo Scientific, Dreieich, Germany) and primers pZ-ASS-F and pZ-ASS-R for pZ-ASS and pKD3-F and pKD3-R for pKD3. The sequence of vectors with correct insert size was confirmed by Sanger sequencing (LGC Genomics, Berlin, DE or Microsynth Seqlab, Göttingen, DE). For in silico sequence analysis, the software Snapgene (GSL Biotech LLC, San Diego, US) or Geneious (Auckland, New Zealand) was used. pBBR1-Lcm was a gift from Thore Rohwerder. Target strains were transformed with 40 ng of the correct plasmids by electroporation in the same manner described for transformations with λ-red knockout cassettes. Successful transformation was confirmed by resistance to plasmid specific antibiotics as well as colony PCR with plasmid specific primers (as described above for pZ-ASS and pKD3; pBBR1-F and pBBR1-R for pBBR1-Lcm) and DreamTaq polymerase (Thermo Scientific, Dreieich, Germany).

## Media and growth experiments
LB medium (1% NaCl, 0.5% yeast extract, 1% tryptone) was used for strain maintenance, cloning and deletion strain construction. If required, antibiotics (kanamycin (50 μg/mL), ampicillin (100 μg/mL), streptomycin, (100 μg/mL), or chloramphenicol (30 μg/mL)) were added. Antibiotics were omitted in growth experiments. Standard M9 minimal media (50 mM $Na_2HPO_4$, 20 mM $KH_2PO_4$, 1 mM NaCl, 20 mM $NH_4Cl$, 2 mM $MgSO_4$ and 100 μM $CaCl_2$, 134 μM EDTA, 13 μM $FeCl_3 \cdot 6H_2O$, 6.2 μM $ZnCl_2$, 0.76 μM $CuCl_2 \cdot 2H_2O$, 0.42 μM $CoCl_2 \cdot 2H_2O$, 1.62 μM $H_3BO_3$, 0.081 μM $MnCl_2 \cdot 4H_2O$) was used for growth experiments and strain evolution with the carbon sources indicated in the text. For growth experiments, precultures were prepared in M9 medium supplemented with and antibiotics for any plasmids present and carbon sources depending on the strain. For strains expressing Lcm, 1 mM IPTG and 1 μM $B_{12}$ (cyanocobalamin) were added to the preculture medium. For the β-alanine aux. and derivative strains with 20 mM glycerol, 10 mM L-lactate and 100 μM 3-HP were added. After harvesting grown overnight cultures (6000 x*g*, 3 min), the cells were washed three times in M9 medium to remove residual carbon sources, antibiotics and cofactors. Growth experiments were performed in 96-well microtiter plates (Nunclon Delta Surface, Thermo Scientific) at 37 °C and were inoculated with washed cells to an optical density ($OD_{600}$) of 0.01 in 150 μL total culture volume per well. To avoid evaporation but allow gas exchange, 50 μL mineral oil (Sigma-Aldrich) were added to each well. If not stated otherwise, growth was monitored in technical triplicates in a BioTek Epoch 2 Microtiterplate reader (BioTek, Bad Friedrichshall, Germany) by absorbance measurements ($OD_{600}$) of each well every ~10 minutes with intermittent orbital and linear shaking. As previously established empirically for the instrument, blank measurements were subtracted and $OD_{600}$ measurements were converted to cuvette $OD_{600}$ values by multiplying with a factor ~4.35. Growth curves represent the average of technical triplicate measurements and were plotted in MATLAB version R2020a.

## Isolation of CoA for ¹³C isotopic labeling analysis
To confirm utilization of the Lcm pathway for growth, isotopic labeling patterns from metabolites of the β-alanine aux. +pBBR1-Lcm +chromosomal-rbsA-*abfT* grown on ¹³$C_3$ L-lactate were determined. To this end, precultures of two biological replicates of the β-alanine aux. strain and a Δ*lldD* strain were first inoculated in M9 + 20 mM glycerol + 1 mM L-lactate + 1 mM IPTG + 1 μM $B_{12}$ and grown at 37 °C to an $OD_{600}$ of > 1. The cells from these precultures were harvested and washed three times in M9 medium as described for growth experiments. The washed

cells were used to inoculate cultures in M9 + 20 mM glycerol + 1 mM ¹³$C_3$ L-lactate + 1 mM IPTG + 1 μM $B_{12}$. Of these, the Δ*lldD* culture was immediately inoculated in technical triplicates, grown to an $OD_{600}$ of 0.8 and immediately harvested. All cultures of β-alanine aux. strains were propagated two more times in M9 + 20 mM glycerol + 1 mM ¹³$C_3$ L-lactate + 1 mM IPTG + 1 μM $B_{12}$, of which the last time technical triplicates were inoculated, which were then harvested at an $OD_{600}$ of 0.8. Harvesting was always performed on ice. In a 2 mL Eppendorf tube, metabolism was quenched by adding 1 mL cell culture of $OD_{600}$ of 0.8 to 1 mL −70 °C cold 70% methanol and inverting the tube once. Cell harvesting was performed at − 10 °C, 10 min, 10,000 x*g*, followed by supernatant removal and pellet storage at −80 °C until endometabolome extraction. For endometabolome extraction, extraction fluid (1:1 mixture of LC-MS grade methanol and TE-buffer pH 7.0 (10 mM TRIZMA, 1 mM EDTA pH neutralized with HCl)) and HPLC grade chloroform were cooled to −20 °C. To each cell pellet 200 μL of −20 °C cold extraction fluid and chloroform per mL of $OD_{600}$ =1 were added. Pellets were resuspended by vortexing prior to shaking incubation at 4 °C for two hours. Phase separation was performed by centrifugation at − 10 °C, 10 min, 10,000 x*g*. The upper phase was carefully removed with a pipette and filtered with a 0.2 μM filter into a new Eppendorf tube stored at −80 °C until metabolite measurement.

## Untargeted ¹³C isotopic labeling analysis for CoA
Quantitative determination of the targets was performed using high-resolution LC-MS. The chromatographic separation was performed as described above for CoA thioester quantification at a constant flow rate of 0.25 mL/min with an injection volume of 5 μL. The mobile phase profile consisted of the following steps and linear gradients: 0–2 min constant at 0% B; 2–5 min from 0 to 6% B; 5–8 min from 6 to 23%; 8–10 min from 23 to 80%; 10–11 min constant at 80% B; 11–12 min from 80 to 0% B; 12 to 18 min constant at 0% B. A Thermo Scientific ID-X Orbitrap mass spectrometer was used in positive mode with an electrospray ionization source and the following conditions: H-ESI spray voltage at 3500 V, sheath gas at 50 arbitrary units, auxiliary gas at 10 arbitrary units, sweep gas at 1 arbitrary unit, ion transfer tube temperature at 325 °C. Detection was performed in full scan mode using the orbitrap mass analyzer at a mass resolution of 240 000 in the mass range 765–790 (*m/z*). Extracted ion chromatograms of the [M + H]+ forms were integrated using Tracefinder software (Thermo Scientific), applying a mass accuracy of 5 ppm. Relative abundance was determined based on the peak area. The used *m/z* values are given in Supplementary Table 4.

## Targeted ¹³C isotopic labeling analysis for CoA
Quantitative determination of the target metabolite was performed as described above for CoA thioester quantification, with the mobile phase profile used for untargeted isotopic labeling analysis. Detection and analysis of the data was performed as described above for CoA thioester quantification. Relative abundance was determined based on the peak area. Mass transitions, collision energies, Cell accelerator voltages and Dwell times have been optimized using chemically pure standards. Parameter settings of all targets are given in Supplementary Table 5.

## Lcm evolution in the β-alanine auxotroph
To employ the eMutaT7 system for Lcm optimization, we cloned the Lcm operon into the target plasmid (pHyo245; referred to pMutaT7 in the main text for simplicity) in a HiFi assembly (New England Biolabs) using the primers o611_pVS133_Lcm-operon_HiFi-F and o612_pVS133_ Lcm-operon_HiFi-R for Lcm amplification from pBBR1-Lcm and o609_pVS133_backb_HiFi-Fwd and o610_pVS133_backb_HiFi-Rev for amplification of the pHyo245-backbone. Successful plasmid construction was verified by plasmid amplification using the primers Ec153_mhp_Pro_V_F and P01_pZ_SF and PrimeStar Max polymerase and subsequent amplicon sequencing by Plasmidsaurus. The β-alanine

auxotrophic strain with a chromosomal rbsA-abfT integration was first electroporated with the pMutaT7-Lcm plasmid, followed by recovery and selection before electroporating with the pMutator plasmid. Two separate cultures of the strain carrying both plasmids were inoculated and grown overnight in LB medium with Ampicillin and Chloramphenicol and 10 mM L-arabinose for induced hypermutation (the workflow is illustrated in Supplementary Fig. 9B). The dense populations were harvested and washed three times in M9 minimal medium, before inoculating cultures with selective medium (20 mM glycerol + 5 mM L-lactate + 1 mM IPTG) in a 1:10 dilution. The selective cultures were grown until an OD > 0.5 was reached before propagating the cells to fresh medium in a 1:10 dilution. After six cultivations in minimal medium, cells were streaked onto LB plates using a 10 μL inoculation loop. Growth of the final populations was compared to that of previous selection strains in a growth experiment. Single colonies were tested for Ampicillin and Chloramphenicol sensitivity. After verifying growth in selective medium for single colonies in tubes (11/15 of library #1 and 3/15 of library #2 grew), their growth was compared to that of previous strains in a plate reader growth experiment. From all grown colonies, the pMutaT7-Lcm plasmid was isolated and amplified using the primers Ec153_mhp_Pro_V_F and P01_pZ_SF and PrimeStar Max polymerase, and the amplicon was sequenced using the Plasmidsaurus amplicon sequencing service. The isolated pMutaT7-Lcm evo plasmids were electroporated into the original β-alanine auxotrophic strain with a chromosomal rbsA-abfT integration again, and growth in selective medium was verified. Mutated Lcm variants were cloned into the pASG-IBA43 vector in a HiFi assembly using the primers pASG-IBA43_fwd and pASG-IBA_rvs for backbone amplification and Lcm_fwd and Lcm_rvs for insert amplification, both with PrimeStar Max.

### Reporting summary
Further information on research design is available in the Nature Portfolio Reporting Summary linked to this article.

## Data availability
Data supporting the findings of this work are available within the paper and its Supplementary Information files. A reporting summary for this Article is available as a Supplementary Information file. The strains reported here are available for academic research upon request from the corresponding authors. The DNA sequences of plasmids constructed and used in this study can be downloaded from the repository EDMOND [https://doi.org/10.17617/3.2GBQWY]. Source data are provided with this paper.

## Code availability
The scripts for the MDF analysis of all acetyl-CoA assimilation pathways are available at Github [https://github.com/helenaschulzmirbach/LCM_MDF].

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

## Acknowledgements

The authors thank Peter Claus for technical support with LC-MS measurements, Andreas Küffner for help with AlphaFold2 modeling of the mutase structures, Daniel Marchal for providing *E. coli* adenylate kinase, Leonard Westphal for pre-tests to establish the MutaT7 system, Jan Zarzycki for providing Pcs and Melanie Klose for technical assistance with protein purifications. The authors are grateful to Thore Rohwerder for gifting multiple plasmids for expression of different Hcm candidates (pBBR1-Lcm, pASG-IBA43-BmHcmA, pET16b-BmHcmB, pASG-IBA43_K-tusRcmA, pASG-IBA43_KtusRcmB). The eMutaT7 target plasmid (referred to as pMutaT7 here) was a gift from Seokhee Kim (pHyo245; Addgene ID 173147). The eMutaT7 mutator plasmid (referred to here as pMutator) was a gift from Seokhee Kim (pDae079; Addgene ID 187622). The authors thank Beau Dronsella, Blake Rasor and Nico Claassens for critical reading of and fruitful discussions on the manuscript. This work was funded by the Max Planck Society. H. S.-M. acknowledges funding by the Bosch Research Foundation and by the Joachim Herz Foundation in form of an Add-On Fellowship for Interdisciplinary Life Science. A.S. acknowledges funding from the International Max Planck Research School for Primary Metabolism and Plant Growth.

## Author contributions

H.S.-M., P.W., and A.S. contributed equally. A.S., A.B.-E., S.B., and T.J.E. conceived the study. A.S., P.W., and H.S.-M. designed the experiments. S.B., P.W., and A.S. identified candidate enzymes in silico. P.W., H.S.-M. performed in vitro experiments with support by M.N. H.S.-M., T.W., H.M., and A.S. constructed strains and performed in vivo experiments. H.S.-M., P.W., and N.P. performed LC-MS analyses. T.J.E. supervised research. H.S.-M., A.S., P.W., and T.J.E. wrote the manuscript with contributions from all authors.

## Funding

## Competing interests

The authors declare no competing interests.
