## [Peer Review file · Nature Communications]

New-to-nature CO₂-dependent acetyl-CoA assimilation enabled by an engineered B₁₂-dependent acyl-CoA mutase

Corresponding Author: Professor Tobias Erb

Version 0:

Reviewer comments:

Reviewer #1

(Remarks to the Author)

General comments

In this study, a coenzyme B₁₂-dependent acyl-CoA mutase was engineered for achieving an improved catalysis of the interconversion of lactyl-CoA and 3-hydroxypropionyl-CoA. The latter reaction is poorly catalyzed by wildtype acyl-CoA mutases, as demonstrated for the 2-hydroxyisobutyryl-CoA mutase subgroup. By combination with other enzymes, a so-called "Lcm module" was established in vitro and (partially/modified) in vivo for the synthesis of 3-hydroxypropionate/lactate from acetyl-CoA and CO₂ via malonyl-CoA, 3-hydroxypropionyl-CoA and lactyl-CoA. In a future economy mainly based on fuels and building block chemicals from renewable carbon, this module might be employed for the implementation of more efficient routes for CO₂ assimilation. Additionally, it provides an interesting pathway for the production of 3-hydroxypropionate, a potential building block for the synthesis of polymers and other chemical products. Overall, the study is very inspiring, employs various state-of-the-art methods for enzyme engineering and characterization and is written in a compact and intelligible way.

I have a couple of more general points regarding importance of the work and interpretation of the results.

- The Lcm module seems to be advantageous, as it is linear, quite short and compatible with aerobic metabolism. And the Lcm module prevents the loss of CO₂ when C₃ compounds have to be synthesized, e.g., for gluconeogenesis. However, is this loss really a great problem (L53 to 55: "...such loss of carbon and energy is not desired")? In an assimilating culture, the CO₂ can be kept in the system, e.g., by recycling the off-gas. So, it depends only on the energy demand for the CO₂ fixation. Unfortunately, NADH/ATP consumption etc. of the various pathways is not compared in this manuscript (see Fig. 1 and Supplementary Figure 1 and 2). And in case of an "energy loss", a decarboxylation reaction might still be advantageous by increasing the driving force of the metabolic route, e.g., synthesis of C₃ compounds.
- In a future bioeconomy, efficient biomass production directly from CO₂ (via C₃ compounds and gluconeogenesis) could be one goal. More important might be the synthesis of fuels (from hydrocarbon precursors >>C₃ chain length) and precursors for polymer production (e.g. carboxylic acids, alcohols and multifunctional molecules; likely not all C₃ compounds).
- The targeted hypermutation and evolution in *E. coli* resulted in a couple of mutase mutants showing improved catalytic efficiency in vitro. In contrast, the performance of the Lcm module did not improve that much in vivo (Fig. 4B) and in vitro (Fig. 5D). As discussed, the in vitro activity might be explained by reduced stability of the mutase mutants. The in vivo performance, however, might be affected by numerous effects not directly related to changes in the mutase sequence (specific cyanocobalamin importer needed, enzymatic conversion of cyanocobalamin to the coenzyme adenosylcobalamin, loading of the mutase with the coenzyme, MeaB-dependent protection and repair mechanism for replacing inactivated coenzyme). These effects might often be more relevant than mutations in the mutase. In line with this, some mutations in the chaperone MeaB are found (Table 1). In conclusion, the applied method seems not to be very efficient for detecting improved mutants (and might even fail to detect them sometimes). Indeed, only a few improving mutations (Table 1) have been detected (after putting a lot of work into it).

Some minor points are also given below (see Specific points). Besides, I found numerous (small) errors and inconsistent use and writing of terms in the manuscript (typos, inconsistent use of abbreviation, in several cases a discrepancy between figure caption/legend and what is shown in the figures etc.). Likely, I did not find all. Therefore, I recommend to thoroughly checking the whole manuscript and the Supplementary materials.

Specific points

L53 and Supplementary Figure 2 using “PEP” versus “phosphoenolpyruvate” in Supplementary Figure 3.

L82, “that were demonstrate recently” change to “...demonstrated...”.

L98, “the conversion of 3-HP-CoA, into lactyl-CoA” Delete comma.

L111, “CoA-ester” I would prefer to write “CoA-thioester”. The term “ester” should only be used when an alcohol group is involved (but in CoA it is a thiol group).

L114, “produced six homologs”: I cannot find any information on the three that did not work. Here, you should also refer to Supplementary Fig. 4A (otherwise it would be not mentioned in the manuscript). Did you test six homologs (L114) or more (Supplementary Fig. 4A might indicate up to 10 mutases)?

L117, “Robertmurray massiliosenegalensis” change to “Robertmurraya massiliosenegalensis”.

L128, “...the enzyme from *A. tertiarycarbonis* had no detectable activity” I would expect some activity with D-lactyl-CoA. Did you test the purified enzyme for activity with 2-hydroxyisobutyryl-CoA? Did you purify an enzyme at least showing the (expected) wildtype activity?

L137 “...while the K_m value for L-lactyl-CoA (0.14 ± 0.09 mM) was comparable to the native substrate 2-hydroxyisobutyryl-CoA...” According to Fig. 2D this is not true. Fig. 2D: $K_m = 0.014$ mM for 2-hydroxyisobutyryl-CoA. By the way, in Fig. 2 caption C and D are mixed up.

L184, “However, rational efforts to target first-shell active site residues rendered the proteins inactive...”. This is really disappointing. And it indicates that we still have a substantial knowledge gap regarding mechanism and function of active site residues in acyl-CoA mutases. On the other hand, inactivation is quite likely, as the radical mechanisms requires a precise positioning of the substrate acyl residue (e.g., the interaction with the hydroxyl group in lactyl- and 3-hydroxypropionyl-CoA is important for correct orientation). A mutation at the active site changing this positioning can cause (irreversible) inactivation of the whole enzyme. Nevertheless, could you provide more details on the criteria for testing these mutations? Models (AlphaFold or homology models) might not be sufficiently reliable to identify all the important residues.

L201, “Lcm (~ 0.02 s⁻¹),...”? See Fig. 2C. So, $0.03/s \pm 0.005/s$ is around 0.02?

Table 1, MeaB mutant “G135G”?????????

L293, “Number of replicates fpr pBBR1-LCM” Correct typo.

L294, “with 1 mM l-lactate” Change to “...L-lactate” Or better change to the R/S nomenclature.

L315 “...variants V921I E137K and P163L compared...” Do you mean “V92I”?

L324, “Outliers omitted in curve-fitting are shown as...” What are your criteria for outliers?

L327, “Variant P163L was inactive and is therefore not shown, it was purified again for further testing” This is confusing. So, the second purification was active? How stable are these enzymes in general? Did you get reproducible activity with the other enzymes/mutants?

Fig. 5D: Although it seems to be indicated (asterisk, “ns”), I cannot really deduce the significance. Maybe, you should provide some details on your test method.

L354, “*E. coli* ArcticExpress”. I guess it is “*E. coli* ArcticExpress (DE3)”. (And “ArcticExpress” should not be in italic type.)

L361, “anhydrotetracycline (pASG-IBA43).” Change to “anhydrotetracycline (pASG-IBA43).”

L361, “Mutase subunits A and B (regardless of source organism) were produced in ArcticExpress, whereas all other proteins were produced in BL21 (DE3).” Might be deleted, as already indicated in L353.

L374 to 393, mutase assay: To the assay, only 12.5 or 15 μ M coenzyme B12 were added. This might be non-optimal for 5 μ M mutase enzyme. To my experience, B12 should be added at much greater excess. Did you test higher concentrations (50, or even 500 μ M)? See also testing of the whole assimilation module (L404).

The assay was performed at 30 °C and pH 7.5. This might be a problem for the *Kyrpidia tusciae* mutase, as it is a thermophilic enzyme with an optimum around 55 °C. Likely, this is the reason for the lower activity (see L127). Did you test higher temperatures than 30 °C for this enzyme?

L408, assimilation module testing: Did you incubate in the presence of oxygen? I have doubts that the mutase could be kept active for several hours (or even 24 hours) at 30 °C in the presence of oxygen and when only adding 12.5 μ M coenzyme B12.

L487, “optimized sequences, see Supplementary.” Add close bracket after “Supplementary”?

L502, "Rhowerder" change to "Rohwerder".

L577 "The chromatographic separation was performed on an...". It looks to me that the LC protocol is identical to the one used for the untargeted ¹³C isotopic labelling analysis. So, L577 to 585 might be shortened by just referring to the above described method.

Supplementary Figure 4: Phylogenetic tree. The B12-binding domain (or subunit) is more conserved among the different mutases. Was this domain included in the analysis?

"...and four uncharacterized B12-dependent mutases were chosen for in vitro testing." I cannot find any information on the enzymes (bacterial strain, accession number for gene/protein sequence) that were chosen but could not be purified (see L114-115).

"3-HP-CoAvia" change to "3-HP-CoA via". "(see panek C)" Do you mean "panel C"?

Supplementary Figure 5: "for the formation of lactyl-CoA formation from lactate" change to "for the formation of lactyl-CoA from lactate".

Supplementary Figure 6: Could you give experimental details, such as initial lactyl-CoA concentration and number of replicates.

Supplementary Figure 7: A) and B) are mixed up in the caption.

Supplementary Figure 8: "growing via the the Lcm pathway" change to "growing via the Lcm pathway".

Supplementary Figure 9: "cheaperone MeaB (dark grey)" change to "chaperone MeaB (dark grey)".

"LB Amp100" and "Cap30" I know what you mean. But this abbreviations are not really consistently used in the manuscript.

Supplementary Figure 10: I do not see the green line = library 2. Instead, there is a purple line that is not explained in the legend of the graph.

Supplementary Figure 12: "None of the obtained mutations ...directly point..." "None" = singular ("points")?

So, none of the mutations is at the active site. And the mutations seem to be mainly at the surface of the protein. Therefore, it might be better to model the natural protein complex that consists of two large and two small subunits. Possibly, there is some improvement in the interactions between these subunits resulting in a better enzyme performance. For the in vivo performance, the interaction with MeaB should be added, as the total protein complex consist of 2 x large subunit, 2 x small subunit and 2 x MeaB).

AlphaFold: Could you provide pLDDT values? Any part of the structure showing low(er) confidence (e.g. some flexible loop region)?

Reviewer #2

(Remarks to the Author)

Authors developed a new metabolic pathway, the Lcm module, in which CO₂ is fixed without the intermediary CO₂ release. By employing the systemic design of a novel metabolic pathway, testing various enzymes for individual steps, and reconstituting the whole pathway in vitro, they confirmed the desired conversion of acetyl-CoA into lactyl-CoA. Furthermore, to improve the bottleneck step, they employed the in vivo targeted hypermutation and selection, identifying a variant with 10-fold improved catalytic efficiency.

This is a solidly developed and well written manuscript. The design principle is novel, and the experiments were clearly described and carefully performed. Together with the previously reported C₁-assimilating pathways, the newly developed Lcm module extends the list of valuable synthetic carbon fixation pathways for future biotechnological applications. This manuscript is of high quality and worthy of publication in Nature Communications. Here are some suggestions authors may consider to improve the readability of the manuscript.

Authors focused on the construction and validation of the pathway only in vitro, and made no mention for the in vivo incorporation of the Lcm module. The latter may require another major efforts for coupling the Lcm module to the cellular metabolic pathways, and therefore, may not be within the scope of this paper. Nevertheless, the additional comments on the future studies involving the Lcm module (e.g. in discussion) will undoubtedly help readers to fully appreciate the significance made in this paper. Also, if any, authors' perspectives about potentials and limitations of the Lcm module compared to others in the in vivo incorporation will be valuable.

In the graphical abstract, the figure describing selection may misleadingly imply that a culture with continuous flow was adopted in this study. A different figure (e.g. which describes serial passaging with dilutions) would prevent any misunderstandings this may cause.

In Supplementary Figure 9B, the dilution factor in selective medium is 1:10. This differs from that stated in the main text (lines 609–610). Either the figure or the text requires an update for the consistency thereof.

line 82, "... that were recently demonstrated,"

Reviewer #3

(Remarks to the Author)

The manuscript "New-to-nature, CO₂-dependent acetyl-CoA assimilation enabled by an engineered B12-dependent acyl-CoA mutase from Schulz-Mirbach et al. presents a novel metabolic pathway, the "Lcm module," which converts acetyl-CoA into pyruvate while assimilating CO₂. An engineered B12-dependent acyl-CoA mutase facilitates this process. The scientific premise is sound, addressing the critical issue of carbon loss in existing aerobic pathways for acetyl-CoA conversion. The approach of using a coenzyme B12-dependent mutase for converting 3-hydroxypropionyl-CoA into lactyl-CoA is innovative and well-justified, given the enzyme's potential for carbon fixation.

The feasibility of the proposed pathway is supported by experimental evidence, including the demonstration of Lcm activity in an engineered enzyme and the enhancement of catalytic efficiency through *in vivo* hypermutation and adaptive evolution. The successful *in vitro* demonstration of the complete Lcm module further substantiates the pathway's practicality. However, this pathway's long-term stability, efficiency under varying conditions, and scalability in an industrial context would require more extensive validation.

Overall, the Technical Approaches employed, including enzyme engineering, hypermutation, and adaptive evolution, are appropriate and align with current synthetic biology and metabolic engineering practices. Using *Bacillus massiliosenegalensis*-derived 2-hydroxyisobutyryl-CoA mutase as a scaffold for engineering the Lcm activity demonstrates a strategic choice of the enzyme, leveraging its structural and functional properties. The results indicating a 10-fold improvement in catalytic efficiency through targeted mutations and adaptive evolution are significant. They demonstrate the potential of the Lcm module to be an effective tool in carbon assimilation and metabolic engineering. The *in vitro* demonstration of the pathway is a critical step in proving the concept. However, *in vivo* integration and performance in a cellular context would be necessary to assess its viability and effectiveness fully.

The authors convincingly argue that the Lcm module represents a valuable addition to the toolkit for metabolic engineering and synthetic biology, with implications for carbon fixation and biomass production. However, the long-term implications for industrial application, including the economic and environmental benefits, need to be explored in future studies.

Overall Assessment

The manuscript presents a novel and scientifically sound approach to acetyl-CoA assimilation that integrates CO₂ fixation, a significant advancement in metabolic engineering. The experimental design and technical execution are robust, supporting the feasibility of the pathway. Future work should focus on integrating this pathway in living organisms and scaling up the process to assess its industrial applicability and sustainability.

As most strength points of this manuscript, I can list the following:

i) The design of an oxygen-tolerant, CO₂-assimilating pathway from acetyl-CoA to pyruvate using the Lcm module addresses a significant metabolic engineering gap; ii) The choice of 2-hydroxyisobutyryl-CoA mutases (Hcm) for developing Lcm activity and the subsequent engineering to improve catalytic efficiency demonstrate thorough methodology and understanding of enzymatic mechanisms; iii) *In Vitro* and *In Vivo* Validation: The study provides both *in vitro* and *in vivo* evidence of the pathway's functionality, enhancing the credibility of the results; iv) A comprehensive analytical approach is illustrated using reaction thermodynamics estimation, enzyme stereospecificity tests, and ¹³C-labelling for metabolic flux analysis.

Some Cons:

The authors should discuss the low catalytic efficiency further: The native and initial engineered versions of the Lcm showed relatively low catalytic activity, which could limit the practical application of the pathway.

There are remaining questions regarding Enzyme Stability Issues: The instability of specific Lcm variants, such as P163L, may affect the long-term viability of the pathway in industrial applications.

Despite many improvements and great potential, the Complexity of the System can be a bottleneck. The pathway involves multiple steps and enzymes, which could complicate the optimization and scaling-up processes.

Some points will need to be improved in future research and should be mentioned in the text, like Enhance Enzyme Stability: Further research could focus on improving the stability of Lcm variants, particularly for those that show high catalytic efficiency but poor stability; Streamline the Pathway: Simplifying the pathway by reducing the number of steps or engineering multifunctional enzymes could improve its efficiency and feasibility for industrial applications and Optimizing the overall pathway to increase pyruvate yield and reduce by-product formation or substrate inhibition could make the process more economically viable.

In summary, while the study presents a promising new pathway for CO₂ assimilation and pyruvate production, further work is needed to address the limitations related to catalytic efficiency, enzyme stability, and system complexity. Additionally, the pathway's long-term practicality and economic viability in industrial applications must be thoroughly evaluated.

Lanes 102-108 - The Lcm module outperforms other pathways with only 7 and 9 reactions for converting acetyl-CoA to pyruvate and oxaloacetate, respectively. Alternative routes lack experimental validation and suffer from oxygen-sensitive,

inefficient biocatalysts. Some quantitative data demonstrating the Lcm module's efficiency would highlight its biotechnological advantages. Predictive analysis could further establish its superiority over existing alternatives.

Line 117- ... *Kyrpidia tusciae* and *Bacillus massiliosenegalensis* (also referred to as *Robertmurray massiliosenegalensis*). Reference?

Discussion:

The provided text functions more as a conclusion than a discussion. It summarizes the study's achievements, highlighting the establishment of the Lcm module, its efficiency improvements, and its potential applications in synthetic biology. The text reflects on the outcomes and implications of the research, offering a succinct overview of the work's significance and future utility, which are characteristics typically found in a conclusion section. A discussion section would delve deeper into analyzing the results, comparing them with existing literature, and exploring the broader implications and limitations of the study.

Some Unresolved Questions could be briefly discussed in a proper discussion section.

-How stable is the engineered pathway in long-term cultures, and does it maintain efficiency under varying environmental and operational conditions?

-What are the potential economic and environmental impacts of scaling up this pathway for industrial use, especially regarding cost-effectiveness and carbon footprint reduction?

-How well can the Lcm module be integrated into different host organisms, especially those used in industrial bioprocessing, without affecting their growth and viability?

I recommend enhancing the discussion throughout the text, as the current discussion section resembles more of a conclusion. It would benefit from a more detailed analysis, comparing the findings with existing literature and exploring the broader implications and limitations of the study, as opposed to merely summarizing the outcomes and potential applications.

The reference list requires meticulous verification to rectify missing data and format discrepancies, such as the improperly formatted reference 40.

Version 1:

Reviewer comments:

Reviewer #1

(Remarks to the Author)

The manuscript has been thoroughly revised and my main points of criticism have been addressed appropriately. From my side, it is almost acceptable for publication. However, there are still some little issues that should be checked.

L147 "None of the mutases showed detectable activity with D-lactyl-CoA, which is in line with previous results demonstrating stereospecificity of these enzymes with their native substrate. 30,38"

Still, this might need some revision, as it not "in line" with the stereospecificity of the *Aquicola* enzyme. Yes, references 30 and 38 deal with the mutases preferring interconversion of R-3-hydroxybutyryl-CoA to 2-hydroxyisobutyryl-CoA. From this, I would also deduce preference for L-lactyl-CoA. However, the enzyme from *Aquicola* clearly prefers S-3-hydroxybutyryl-CoA, as the hydroxyl group is interacting with D117 via H-bonds (see PDB 4R3U, reference 31). Due to this, D-lactyl-CoA (R-lactyl-CoA) should be clearly favored.

L170 "The enzyme activity with L-lactyl-CoA was even slower ($k_{cat} = 0.03 \pm 0.01 \text{ s}^{-1}$), while the K_m value for L-lactyl-CoA ($0.14 \pm 0.09 \text{ mM}$) was higher than for the native substrate 2-hydroxyisobutyryl-CoA (Fig. 2C, D)." Yes, the rate is lower (10 times) and the K_m is higher (also 10 times). That means that both parameters (k_{cat} and K_m) are worse than with the native substrates = the efficiency is 100 times lower. Therefore, maybe, the use of "while" is somewhat misleading (when understood in the sense of "whereas" and indicating a contrast).

" k_{cat} ": use italic type throughout (e.g. L503).

Supplementary Figure 4:

"via the LCM over time (see panel C)." It is panel D now.

"wild-type" (e.g. L90) versus "wild type" (e.g. L286) versus "wildtype" (e.g. L90Supplementary Figure 14).

Obviously, there are still some inconsistencies of writing in the manuscript.

Reviewer #2

(Remarks to the Author)

Authors carefully addressed all the comments with more data and discussions, which further clarify the advances the authors have made in this manuscript. Now this reviewer recommends publication.

Reviewer #3

(Remarks to the Author)

The study introduces a new CO₂-assimilating pathway—the Lcm module—which efficiently transforms acetyl-CoA into pyruvate without carbon loss, offering a faster, oxygen-tolerant alternative to traditional methods. The authors have done an excellent job revising the manuscript, thoughtfully incorporating both my suggestions and the recommendations of other reviewers. Their clear and convincing explanations for the changes made, or not made, leave me satisfied with their revisions.

Point-to-point reply

Reviewer #1 (Remarks to the Author):

General comments

In this study, a coenzyme B12-dependent acyl-CoA mutase was engineered for achieving an improved catalysis of the interconversion of lactyl-CoA and 3-hydroxypropionyl-CoA. The latter reaction is poorly catalyzed by wildtype acyl-CoA mutases, as demonstrated for the 2-hydroxyisobutyryl-CoA mutase subgroup. By combination with other enzymes, a so-called "Lcm module" was established in vitro and (partially/modified) in vivo for the synthesis of 3-hydroxypropionate/lactate from acetyl-CoA and CO₂ via malonyl-CoA, 3-hydroxypropionyl-CoA and lactyl-CoA. In a future economy mainly based on fuels and building block chemicals from renewable carbon, this module might be employed for the implementation of more efficient routes for CO₂ assimilation. Additionally, it provides an interesting pathway for the production of 3-hydroxypropionate, a potential building block for the synthesis of polymers and other chemical products. Overall, the study is very inspiring, employs various state-of-the-art methods for enzyme engineering and characterization and is written in a compact and intelligible way.

We thank the reviewer for the positive remarks and appreciate his/her thoughtful comments and suggestions, which helped us to improve the manuscript.

I have a couple of more general points regarding importance of the work and interpretation of the results.

- The Lcm module seems to be advantageous, as it is linear, quite short and compatible with aerobic metabolism. And the Lcm module prevents the loss of CO₂ when C₃ compounds have to be synthesized, e.g., for gluconeogenesis. However, is this loss really a great problem (L53 to 55: "...such loss of carbon and energy is not desired")? In an assimilating culture, the CO₂ can be kept in the system, e.g., by recycling the off-gas. So, it depends only on the energy demand for the CO₂ fixation. Unfortunately, NADH/ATP consumption etc. of the various pathways is not compared in this manuscript (see Fig. 1 and Supplementary Figure 1 and 2). And in case of an "energy loss", a decarboxylation reaction might still be advantageous by increasing the driving force of the metabolic route, e.g., synthesis of C₃ compounds.

We agree with the points raised by the reviewer and provide new data that compares the different pathways not only in respect to carbon atom efficiency, but also in respect to **thermodynamic favorability** and **resource consumption** (Supplementary Data 1, Supplementary Figures 1 and 2):

We also amended the main text that summarizes these findings by the following statement:

"The Lcm module outperforms most other acetyl-CoA assimilation routes in terms of thermodynamic favorability (max-min driving force, MDF)²³⁻²⁵ and ATP requirement (Supplementary Data 1, Supplementary Fig. 2)."

- In a future bioeconomy, efficient biomass production directly from CO₂ (via C₃ compounds and gluconeogenesis) could be one goal. More important might be the synthesis of fuels (from hydrocarbon precursors >>C₃ chain length) and precursors for polymer production (e.g. carboxylic acids, alcohols and multifunctional molecules; likely not all C₃ compounds).

We agree with the reviewer that biomass and C₃ compounds are not the only desired products for future industrial applications. However, various prospective biofuels and biopolymers can be produced from C₃ compounds such as pyruvate, including isobutanol, isobutylene and polymers based on lactic acid (see Lee et al. *Nat Catal* 2019).

Also, as pointed out by the reviewer, C3 compounds are required for cellular growth, which will be crucial in most microbial bioproductions (especially continuous bioproduction). Any metabolic inefficiency lowering the synthesis of biomass precursors will likely translate to limited overall yields or productivities, thus also affecting the yields of desired non-biomass products. We re-phrased the paragraph as follows to make this point clearer:

“However, to synthesize C3 intermediates of central metabolism, such as pyruvate or phosphoenolpyruvate (PEP), oxaloacetate needs to be decarboxylated, which leads to a loss of previously fixed carbon. In the context of microbial CO₂ valorization, such loss of carbon is not desired, as it requires additional investment of energy and other cellular resources for (re-)assimilation of the released CO₂.”

- The targeted hypermutation and evolution in *E. coli* resulted in a couple of mutase mutants showing improved catalytic efficiency *in vitro*. In contrast, the performance of the Lcm module did not improve that much *in vivo* (Fig. 4B) and *in vitro* (Fig. 5D). As discussed, the *in vitro* activity might be explained by reduced stability of the mutase mutants. The *in vivo* performance, however, might be affected by numerous effects not directly related to changes in the mutase sequence (specific cyanocobalamin importer needed, enzymatic conversion of cyanocobalamin to the coenzyme adenosylcobalamin, loading of the mutase with the coenzyme, MeaB-dependent protection and repair mechanism for replacing inactivated coenzyme). These effects might often be more relevant than mutations in the mutase. In line with this, some mutations in the chaperone MeaB are found (Table 1). In conclusion, the applied method seems not to be very efficient for detecting improved mutants (and might even fail to detect them sometimes). Indeed, only a few improving mutations (Table 1) have been detected (after putting a lot of work into it).

We agree with the reviewer that several other cellular processes/components could be further optimized to support functioning of the Lcm module in *E. coli*. However, we would argue that it is unlikely that the import of (cyano)cobalamin or its conversion to adenosylcobalamin are currently rate-limiting growth, because *E. coli* can sustain much higher flux via native B₁₂-dependent pathways, compared to the requirements of our selection scheme. For instance, *E. coli* can utilize ethanolamine as the sole source of carbon and energy, where all cellular carbon passes through a B₁₂-dependent enzyme (ethanolamine ammonia-lyase; Escalante-Semarens et al. *EcoSal Plus* 2008). However, as the reviewer points out, we cannot exclude that loading of the mutase with coenzyme B₁₂ or MeaB activity, respectively, might be rate-limiting. In fact, this was why we included the *meaB* gene in the hypermutation target sequence, yet found only one synonymous MeaB mutation in our evolution.

We agree with the reviewer that an untargeted, global mutagenesis/evolution approach could be additionally pursued in the future to further optimize other factors, besides MeaB and the Lcm enzyme. However, we consider this to be outside the scope of this work, which really aimed to introduce the proposed pathway and demonstrate the *in vitro* and *in vivo* feasibility of the reaction sequence.

Some minor points are also given below (see Specific points). Besides, I found numerous (small) errors and inconsistent use and writing of terms in the manuscript (typos, inconsistent use of abbreviation, in several cases a discrepancy between figure caption/legend and what is shown in the figures etc.). Likely, I did not find all. Therefore, I recommend to thoroughly checking the whole manuscript and the Supplementary materials.

We apologize for these mistakes, corrected them, and carefully searched the manuscript for additional typos.

Specific points

L53 and Supplementary Figure 2 using “PEP” versus “phosphoenolpyruvate” in Supplementary Figure 3.

We unified the representation in the manuscript.

L82, “that were demonstrate recently” change to “...demonstrated...”.

We corrected the typo.

L98, “the conversion of 3-HP-CoA, into lactyl-CoA” Delete comma.

We corrected the typo.

L111, “CoA-ester” I would prefer to write “CoA-thioester”. The term “ester” should only be used when an alcohol group is involved (but in CoA it is a thiol group).

We agree and modified the relevant sections accordingly.

L114, “produced six homologs”: I cannot find any information on the three that did not work. Here, you should also refer to Supplementary Fig. 4A (otherwise it would be not mentioned in the manuscript). Did you test six homologs (L114) or more (Supplementary Fig. 4A might indicate up to 10 mutases)?

We wanted to test six homologs but could only purify the three that had previously been characterized. We now list in Supplementary Figure 4A the mutases, which we failed to purify, and those that we tested for activity with lactyl-CoA. We modified the main text to explain this more clearly:

“We heterologously produced six homologs (Supplementary Fig. 4A), of which we could successfully purify only the three candidates that had already been characterized previously^{29,30,34}, namely the 2-hydroxyisobutyryl-CoA mutases from Aquincola tertiaricarbonis, Kyrpidia tusciae and Bacillus massiliosenegalensis (also referred to as Robertmurraya massiliosenegalensis)^{35–37}.”

L117, “Robertmurray massiliosenegalensis” change to “Robertmurraya massiliosenegalensis”.

We corrected the typo.

L128, “...the enzyme from A. tertiaricarbonis had no detectable activity” I would expect some activity with D-lactyl-CoA. Did you test the purified enzyme for activity with 2-hydroxyisobutyryl-CoA? Did you purify an enzyme at least showing the (expected) wildtype activity?

The mutase from A. tertiaricarbonis had no activity with 1 mM D-lactyl-CoA in our assays. To confirm that the enzyme was generally active, we spiked the assay with 1 mM 2-hydroxyisobutyryl-CoA, for which we saw a slope corresponding to a turnover of 1.5 min⁻¹. We now added these results as Supplementary Figure 4C and refer to them in the main text:

“In contrast, the enzyme candidate from A. tertiaricarbonis had no detectable activity with L-lactyl-CoA (data not shown), despite the fact that we confirmed functionality of this purified enzyme with its native substrate 2-hydroxyisobutyryl-CoA (Supplementary Fig. 4C). None of the mutases showed detectable activity with D-lactyl-CoA, which is in line with previous results demonstrating stereospecificity of these enzymes with their native substrate^{30,38}.”

L137 "...while the K_m value for L-lactyl-CoA (0.14 ± 0.09 mM) was comparable to the native substrate 2-hydroxyisobutyryl-CoA..." According to Fig. 2D this is not true. Fig. 2D: $K_m = 0.014$ mM for 2-hydroxyisobutyryl-CoA. By the way, in Fig. 2 caption C and D are mixed up.

We thank the reviewer for pointing out these errors and corrected the relevant sections.

L184, "However, rational efforts to target first-shell active site residues rendered the proteins inactive...". This is really disappointing. And it indicates that we still have a substantial knowledge gap regarding mechanism and function of active site residues in acyl-CoA mutases. On the other hand, inactivation is quite likely, as the radical mechanisms requires a precise positioning of the substrate acyl residue (e.g., the interaction with the hydroxyl group in lactyl- and 3-hydroxypropionyl-CoA is important for correct orientation). A mutation at the active site changing this positioning can cause (irreversible) inactivation of the whole enzyme. Nevertheless, could you provide more details on the criteria for testing these mutations? Models (AlphaFold or homology models) might not be sufficiently reliable to identify all the important residues.

We fully agree that mutases are very difficult enzymes to engineer, as they handle radical intermediates, which are hard to control and steer in reactivity. We thank the reviewer for pointing this out and added Supplementary Figure 6A to explain in more detail, which residues we chose as mutagenesis targets. In the figure legend, we describe, how we identified the targets for mutagenesis, as follows:

"Single amino acid mutations were introduced with the intent of narrowing the active site to improve the activity with lactyl-CoA. We chose residues that we expected to surround the acyl-function of the CoA thioester with about 5 Å distance according to the AlphaFold prediction of the Lcm structure (colored in blue, the distance to 2-HIB-CoA in an AlphaFold2 model of the Lcm is indicated in yellow). With this, we aimed to restrict the space available so lactyl-CoA could bind better. We determined CoA thioester binding sites by using the published crystal structure of the A. tertiaricarbonis homologue as template (PDB: 4r3u). The B₁₂ cofactor is colored magenta."

L201, "Lcm (~ 0.02 s⁻¹),..."? See Fig. 2C. So, $0.03/s \pm 0.005/s$ is around 0.02?

We thank the reviewer for pointing this out. We corrected the mistake in L201, which now reads "Because of the low catalytic activity of Lcm (0.03 s⁻¹), we aimed to..."

Table 1, MeaB mutant "G135G"?????????

We obtained one mutant with a silent/synonymous mutation in MeaB (which may result in changed expression level or folding, due to differential codon usage). Notably, when we re-transformed the unevolved β -alanine auxotrophic strain with this plasmid (evo3), it did not grow (Fig. 4B). We clarified the silent mutation in Table 1 by writing "silent G135G (tgg \rightarrow ggg)".

L293, "Number of replicates fpr pBBR1-LCM" Correct typo.

We corrected the typo.

L294, "with 1 mM l-lactate" Change to "...L-lactate" Or better change to the R/S nomenclature.

We corrected the typo but prefer to remain with the L/D, as the field typically uses L/D nomenclature.

L315 "...variants V921I E137K and P163L compared..." Do you mean "V92I"?

We thank the reviewer for pointing out this typo, we indeed meant V92I and corrected that in the text.

L324, “Outliers omitted in curve-fitting are shown as...” What are your criteria for outliers?

We thank the reviewer for pointing this out. We amended the legend of Figure 5, as follows:

“Outliers (defined as values that differed by >15% from the mean; shown as empty symbols) were excluded for curve-fitting.”

L327, “Variant P163L was inactive and is therefore not shown, it was purified again for further testing” This is confusing. So, the second purification was active? How stable are these enzymes in general? Did you get reproducible activity with the other enzymes/mutants?

The P163L variant was the only variant that seemed less stable and was therefore more difficult to purify (or keep) in an active state. The P163L variant quickly lost activity upon prolonged incubation on ice. Of three individual purifications performed in this study, one was completely inactive. In general, loss of enzyme stability upon mutation of the substrate binding pocket, as in P163L, is a commonly observed phenomenon. However, since other variants are anyway more active, we conclude that these are more interesting for future applications. We added a section to the discussion which mentions stability as a point for improvement in the future:

“For future applications in a cell-free context, optimization of the stability and oxygen-tolerance of Lcm may represent another important target, since inactivation of B₁₂-dependent mutases by molecular oxygen is well-described, especially in vitro^{71–76}. We emphasize that applications of Lcm in vivo allow the co-expression of chaperones (MeaB)^{71,77}, salvage of damaged coenzyme B₁₂⁷⁸ and improved protection of the coenzyme from photolysis and oxidative damage in the reductive intracellular environment. Applications in vivo thus likely present a more promising environment for future implementations of Lcm compared to in vitro systems.”

Fig. 5D: Although it seems to be indicated (asterisk, “ns”), I cannot really deduce the significance. Maybe, you should provide some details on your test method.

We added the relevant information to the caption of Figure 5 and changed the representation from asterisks to displaying p-values in the figure.

L354, “E. coli ArcticExpress”. I guess it is “E. coli ArcticExpress (DE3)”. (And “ArcticExpress” should not be in italic type.)

We thank the reviewer for pointing this out. We corrected it.

L361, “anhydrotetracycline (pASG-IBA43).” Change to “anhydrotetracycline (pASG-IBA43).”

We thank the reviewer for pointing this out. We corrected it.

L361, “Mutase subunits A and B (regardless of source organism) were produced in ArcticExpress, whereas all other proteins were produced in BL21(DE3).” Might be deleted, as already indicated in L353.

We thank the reviewer for pointing this out. We corrected it and deleted the sentence in L397.

L374 to 393, mutase assay: To the assay, only 12.5 or 15 μM coenzyme B12 were added. This might be non-optimal for 5 μM mutase enzyme. To my experience, B12 should be added at much greater excess. Did you test higher concentrations (50, or even 500 μM)? See also testing of the whole assimilation module (L404).

As suggested by the reviewer, we also tested the isolated Lcm reaction (separate from the cascade) with a large excess of B12 and refer to this experiment in the main text:

“Given the sensitivity of B₁₂-dependent enzymes to inactivation by molecular oxygen, particularly in vitro, we hoped that supplying more B₁₂ could improve Lcm activity and thus the lactyl-CoA yield of the Lcm module in our in vitro assay. To test this hypothesis, we first compared the mutase turnover with 12.5 μM B₁₂ (2.5x excess over enzyme) and 500 μM (100x excess), and found that higher B₁₂ concentration increased enzyme turnover by 30 % (Supplementary Fig. 14A).”

The assay was performed at 30 °C and pH 7.5. This might be a problem for the *Kyrpidia tusciae* mutase, as it is a thermophilic enzyme with an optimum around 55 °C. Likely, this is the reason for the lower activity (see L127). Did you test higher temperatures than 30 °C for this enzyme?

We thank the reviewer for the comment. We did not test higher temperatures for the enzyme, because we aimed to apply the mutase in mesophilic environments (*E. coli*). To point that out more clearly, we re-phrased the relevant section, as follows:

*“The enzyme from the mesophilic organism *B. massiliosenegalensis* showed the highest activity with L-lactyl-CoA at 30°C (Fig. 2C), while the mutase from the thermophilic organism *K. tusciae* had about 2-fold lower activity in these conditions. In contrast, the enzyme candidate from *A. tertiaricarbonis* had no detectable activity with L-lactyl-CoA (data not shown), despite the fact that we confirmed functionality of this purified enzyme with its native substrate 2-hydroxyisobutyryl-CoA (Supplementary Fig. 4C). None of the mutases showed detectable activity with D-lactyl-CoA, which is in line with previous results demonstrating stereospecificity of these enzymes with their native substrate^{30,38}.*

*Due to its mesophilic properties and the basal activity with L-lactyl-CoA, we decided to further test the Hcm from *B. massiliosenegalensis* and confirmed that the enzyme also catalyzed lactyl-CoA formation from 3-HP-CoA (Supplementary Fig. 4D-F) and determined the kinetic parameters for both the native and promiscuous reaction, forming 3-hydroxybutyryl-CoA or 3-HP-CoA, respectively (Fig. 2C, D).”*

L408, assimilation module testing: Did you incubate in the presence of oxygen? I have doubts that the mutase could be kept active for several hours (or even 24 hours) at 30 °C in the presence of oxygen and when only adding 12.5 μM coenzyme B12.

Yes, we incubated the reaction mixtures in the presence of oxygen. We agree that for incubations over longer time frames, the amount of B12 might limit enzyme catalysis. However, our group has observed in previous studies on two different pathways that excess concentrations of B12 can be inhibitory to *in vitro* enzyme cascades (Pandi et al. *Nat Commun*, 2022); Luo et al. *Nat Catal* 2023). We hypothesized that this may be due to inhibitory effects of cobalt released from damaged B12.

We note in the Discussion that enzyme and coenzyme stability is less of an issue for *in vivo* implementation (which was the primary goal of our study), where cells can degrade, salvage and/or re-synthesize damaged mutase and/or B12. Inside cells, B12 is generally more protected from oxidative damage by the reductive environment, and the activity of specific chaperones / exchange factors, like MeaB, protects Lcm from inactivation.

Nonetheless, we tested the Lcm module (enzyme cascade) with higher concentrations of B12 and describe the results in the main text:

“However, when we tested the entire cascade with 12.5 μM (2.5x excess over enzyme), 100 μM (20x excess) and 500 μM B12 (100x excess), the B12-dependent increase in lactyl-CoA concentrations after six hours was much less pronounced (Supplementary Fig. 14B). Since high B12-concentrations were previously shown to impair in vitro operation of the CETCH and THETA cycle^{9,50}, we hypothesized that one of the other Lcm module enzymes might be inhibited by the additional B12. Indeed, we saw that Pcs dependent 3-HP-CoA formation from 3-HP decreased with increasing B12 concentrations (Supplementary Fig. 14B). To test whether Pcs would suffer from B12-

dependent inhibition, we tested only Pcs and wildtype Lcm on 1 mM 3-HP and found a severe decrease in 3-HP-CoA formation that was correlated to the B12 concentration (Supplementary Fig. 14C). In summary, these findings indicate that B12 stability and complex interactions of the cofactor with the reaction cascade is a major drawback in vitro that would need to be further engineered in the future.”

L487, “optimized sequences, see Supplementary.” Add close bracket after “Supplementary”?

We added the missing bracket.

L502, “Rhowerder” change to “Rohwerder”.

We corrected it.

L577 “The chromatographic separation was performed on an...”. It looks to me that the LC protocol is identical to the one used for the untargeted 13C isotopic labelling analysis. So, L577 to 585 might be shortened by just referring to the above described method.

We thank the reviewer for the comment and absolutely agree. We shortened the corresponding section accordingly.

Supplementary Figure 4: Phylogenetic tree. The B12-binding domain (or subunit) is more conserved among the different mutases. Was this domain included in the analysis? “...and four uncharacterized B12-dependent mutases were chosen for in vitro testing.” I cannot find any information on the enzymes (bacterial strain, accession number for gene/protein sequence) that were chosen but could not be purified (see L114-115).

We thank the reviewer for the comment. We added in the figure caption that the large subunit was used for phylogenetic analysis, and also added the NCBI identifiers for the subunits we could not purify:

“The large subunits (A subunit) of the following putative mutases could not be purified (red exclamation mark): HBB18459.1 (SsHcmA), HBB18460.1 (SsHcmB), HIJ39843.1 (DpHcmA), HIJ39844.1 (DpHcmB), MAF34037.1 (DsHcmA) and MAF34038.1 (DsHcmB).”

“3-HP-CoAvia” change to “3-HP-CoA via”. “(see panek C)” Do you mean “panel C”?

We corrected the typos.

Supplementary Figure 5: “for the formation of lactyl-CoA formation from lactate” change to “for the formation of lactyl-CoA from lactate”.

We corrected the duplication.

Supplementary Figure 6: Could you give experimental details, such as initial lactyl-CoA concentration and number of replicates.

We thank the reviewer for the comment and added the experimental details to the figure caption:

“1.5 mM L-lactyl-CoA was used with 5 μ M of each subunit with 12.5 μ M of coenzyme B12 at 30°C (n=2).”

Supplementary Figure 7: A) and B) are mixed up in the caption.

We thank the reviewer for the comment and corrected the caption.

Supplementary Figure 8: “growing via the the Lcm pathway” change to “growing via the Lcm pathway”.

We removed the duplicate word.

Supplementary Figure 9: “cheaperone MeaB (dark grey)” change to “chaperone MeaB (dark grey)”.

“LB Amp100” and “Cap30” I know what you mean. But this abbreviations are not really consistently used in the manuscript.

We corrected the typo and wrote the full name of the antibiotic instead of abbreviations.

Supplementary Figure 10: I do not see the green line = library 2. Instead, there is a purple line that is not explained in the legend of the graph.

We corrected the color scheme in the legend.

Supplementary Figure 12: “None of the obtained mutations ...directly point...” “None” = singular (“points”)?

We corrected the typo.

So, none of the mutations is at the active site. And the mutations seem to be mainly at the surface of the protein. Therefore, it might be better to model the natural protein complex that consists of two large and two small subunits. Possibly, there is some improvement in the interactions between these subunits resulting in a better enzyme performance. For the in vivo performance, the interaction with MeaB should be added, as the total protein complex consist of 2 x large subunit, 2 x small subunit and 2 x MeaB). AlphaFold: Could you provide pLDDT values? Any part of the structure showing low(er) confidence (e.g. some flexible loop region)?

We thank the reviewer for the comment. Indeed, the mutations are located on the surface of the protein. Following the reviewer’s suggestion to evaluate the mutation positions further, we now modeled the multimer structure with “AlphaFold Multimer” (based on AlphaFold2) using the stoichiometry described by the reviewer, which matches an already published Hcm structure (PDB 4r3u, dimer of dimers, plus two MeaB chaperones). We have added these results as Supplementary Figure 13.

The mutated residues we found in the evolution are not located at the interface of the heterodimers, and neither on flexible loops with low prediction confidence. We have visualized this in the new Supplementary Figure 13 and hope that the figure better illustrates the position of the mutations and appropriately visualizes pLDDT/confidence values.

Reviewer #2 (Remarks to the Author):

Authors developed a new metabolic pathway, the Lcm module, in which CO₂ is fixed without the intermediary CO₂ release. By employing the systemic design of a novel metabolic pathway, testing various enzymes for individual steps, and reconstituting the whole pathway in vitro, they confirmed the desired conversion of acetyl-CoA into lactyl-CoA. Furthermore, to improve the bottleneck step, they employed the in vivo targeted hypermutation and selection, identifying a variant with 10-fold improved catalytic efficiency.

This is a solidly developed and well written manuscript. The design principle is novel, and the experiments were clearly described and carefully performed. Together with the previously reported C1-assimilating pathways, the newly developed Lcm module extends the list of valuable synthetic carbon fixation pathways for future biotechnological applications. This manuscript is of high quality and worthy of publication in Nature Communications. Here are some suggestions authors may consider to improve the readability of the manuscript.

We thank the reviewer for the positive feedback.

Authors focused on the construction and validation of the pathway only in vitro, and made no mention for the in vivo incorporation of the Lcm module. The latter may require another major efforts for coupling the Lcm module to the cellular metabolic pathways, and therefore, may not be within the scope of this paper. Nevertheless, the additional comments on the future studies involving the Lcm module (e.g. in discussion) will undoubtedly help readers to fully appreciate the significance made in this paper. Also, if any, authors' perspectives about potentials and limitations of the Lcm module compared to others in the in vivo incorporation will be valuable.

Currently, the low turnover of the mutase is likely the bottleneck for the implementation in more demanding selections. Following this reviewer comment, we now elaborate more on future directions and limitations in the Discussion:

*“However, for the majority of these applications, the Lcm activity must be further improved. Both acetyl-CoA assimilation and synthetic CO₂ fixation cycles require high flux. Despite the significant performance improvement compared to the wild-type Lcm, the turnover of the best Lcm mutant described here ($k_{cat} = 0.11 \pm 0.009 \text{ s}^{-1}$) is still two orders of magnitude below that of an “average enzyme” ($k_{cat} \approx 10 \text{ s}^{-1}$)³⁹. Thus, all future applications would benefit from Lcm variants with higher turnover than achieved so far. We envision that such improvements could be achieved by additional rounds of mutagenesis and subsequent screening using the workflows and selections established here, or in selections with higher flux demand. Since the Hcm scaffold we chose is slow even with its native substrate, we expect that sampling further candidates from the natural diversity of Hcm homologues may aid the identification of faster Lcm catalysts^{29,32,33,69}. Indeed, related B₁₂-dependent mutases achieve much higher turnover frequencies. For instance, methylmalonyl-CoA mutase (Mcm) is part of central metabolism in multiple organisms and has been reported to achieve k_{cat} values above 200 s^{-1} (e.g. *Methylobacterium extorquens* Mcm with a k_{cat} of 255 s^{-1})^{70,71}. Therefore, we reason that the reaction mechanism itself should allow evolving the lactyl-CoA mutase towards even higher turnovers than those observed for the native reaction of Hcm^{29,32,33}.”*

In the graphical abstract, the figure describing selection may misleadingly imply that a culture with continuous flow was adopted in this study. A different figure (e.g. which describes serial passaging with dilutions) would prevent any misunderstandings this may cause.

We agree with the reviewer and modified the graphical abstract accordingly to illustrate serial dilutions rather than a continuous-flow bioreactor.

In Supplementary Figure 9B, the dilution factor in selective medium is 1:10. This differs from

that stated in the main text (lines 609–610). Either the figure or the text requires an update for the consistency thereof.

We thank the reviewer for pointing out this inconsistency and corrected the information in the text.

line 82, "... that were recently demonstrated,"

We corrected the typo.

Reviewer #3 (Remarks to the Author):

The manuscript "New-to-nature, CO₂-dependent acetyl-CoA assimilation enabled by an engineered B12-dependent acyl-CoA mutase from Schulz-Mirbach et al. presents a novel metabolic pathway, the "Lcm module," which converts acetyl-CoA into pyruvate while assimilating CO₂. An engineered B12-dependent acyl-CoA mutase facilitates this process. The scientific premise is sound, addressing the critical issue of carbon loss in existing aerobic pathways for acetyl-CoA conversion. The approach of using a coenzyme B12-dependent mutase for converting 3-hydroxypropionyl-CoA into lactyl-CoA is innovative and well-justified, given the enzyme's potential for carbon fixation.

The feasibility of the proposed pathway is supported by experimental evidence, including the demonstration of Lcm activity in an engineered enzyme and the enhancement of catalytic efficiency through *in vivo* hypermutation and adaptive evolution. The successful *in vitro* demonstration of the complete Lcm module further substantiates the pathway's practicality. However, this pathway's long-term stability, efficiency under varying conditions, and scalability in an industrial context would require more extensive validation.

We thank the reviewer for the positive assessment. As response to the comment, we expanded the discussion section to discuss current limitations and future efforts to improve the Lcm module (see below).

Overall, the Technical Approaches employed, including enzyme engineering, hypermutation, and adaptive evolution, are appropriate and align with current synthetic biology and metabolic engineering practices. Using *Bacillus massiliosenegalensis*-derived 2-hydroxyisobutyryl-CoA mutase as a scaffold for engineering the Lcm activity demonstrates a strategic choice of the enzyme, leveraging its structural and functional properties. The results indicating a 10-fold improvement in catalytic efficiency through targeted mutations and adaptive evolution are significant. They demonstrate the potential of the Lcm module to be an effective tool in carbon assimilation and metabolic engineering. The *in vitro* demonstration of the pathway is a critical step in proving the concept. However, *in vivo* integration and performance in a cellular context would be necessary to assess its viability and effectiveness fully.

We thank the reviewer for appreciating our experimental strategy. We agree that an *in vivo* implementation of the **complete** Lcm module in the direction of acetyl-CoA conversion to pyruvate inside a growing cell will be an important future step to demonstrate (and optimize/evolve) the effectiveness of this pathway. This is an ongoing effort, which, however, we consider to be outside the scope of this work. We believe that our current study lays the groundwork for this future work by providing several proofs-of-concept (i.e., design and theoretical analyses of the pathway, *in vitro* confirmation of the reaction sequence, as well as *in vivo* demonstration of all key activities of the Lcm module in the direction of lactate conversion to 3-hydroxypropionate).

The authors convincingly argue that the Lcm module represents a valuable addition to the toolkit for metabolic engineering and synthetic biology, with implications for carbon fixation and biomass production. However, the long-term implications for industrial application, including the economic and environmental benefits, need to be explored in future studies.

We agree with the reviewer that future studies on the industrial applicability and utility of this pathway should incorporate techno-economic analyses.

Overall Assessment

The manuscript presents a novel and scientifically sound approach to acetyl-CoA assimilation that integrates CO₂ fixation, a significant advancement in metabolic engineering. The experimental design and technical execution are robust, supporting the feasibility of the pathway. Future work should focus on integrating this pathway in living organisms and scaling up the process to assess its industrial applicability and sustainability.

As most strength points of this manuscript, I can list the following: i) The design of an oxygen-tolerant, CO₂-assimilating pathway from acetyl-CoA to pyruvate using the Lcm module addresses a significant metabolic engineering gap; ii) The choice of 2-hydroxyisobutyryl-CoA mutases (Hcm) for developing Lcm activity and the subsequent engineering to improve catalytic efficiency demonstrate thorough methodology and understanding of enzymatic mechanisms; iii) In Vitro and In Vivo Validation: The study provides both in vitro and in vivo evidence of the pathway's functionality, enhancing the credibility of the results; iv) A comprehensive analytical approach is illustrated using reaction thermodynamics estimation, enzyme stereospecificity tests, and ¹³C-labelling for metabolic flux analysis.

We thank the reviewer for highlighting these aspects of our manuscript.

Some Cons:

The authors should discuss the low catalytic efficiency further: The native and initial engineered versions of the Lcm showed relatively low catalytic activity, which could limit the practical application of the pathway.

There are remaining questions regarding Enzyme Stability Issues: The instability of specific Lcm variants, such as P163L, may affect the long-term viability of the pathway in industrial applications.

We thank the reviewer for their constructive criticism. We agree that wild-type Lcm and Lcm variants show relatively low catalytic activity for the non-native reaction. As the reviewer notes correctly, enzyme stability issues could indeed cause challenges for certain future applications, particularly in a cell-free, *in vitro* context (please also see our answers to reviewer #1).

Following these comments and suggestions by reviewer #1 and #3, we expanded the Discussion section to discuss current limitations and necessary future efforts to improve the Lcm module such as stability and turnover rates:

*“However, for the majority of these applications, the Lcm activity must be further improved. Both acetyl-CoA assimilation and synthetic CO₂ fixation cycles require high flux. Despite the significant performance improvement compared to the wild-type Lcm, the turnover of the best Lcm mutant described here ($k_{cat} = 0.11 \pm 0.009 \text{ s}^{-1}$) is still two orders of magnitude below that of an “average enzyme” ($k_{cat} \approx 10 \text{ s}^{-1}$)³⁹. Thus, all future applications would benefit from Lcm variants with higher turnover than achieved so far. We envision that such improvements could be achieved by additional rounds of mutagenesis and subsequent screening using the workflows and selections established here, or in selections with higher flux demand. Since the Hcm scaffold we chose is slow, even with its native substrate, we expect that sampling further candidates from the natural diversity of Hcm homologues may aid the identification of faster Lcm catalysts^{29,32,33,69}. Indeed, B₁₂-dependent mutases can in principle achieve much higher turnover numbers. For instance, methylmalonyl-CoA mutase (Mcm) is part of central metabolism in multiple organisms and has been reported to achieve k_{cat} values above 200 s^{-1} (e.g. *Methylobacterium extorquens* Mcm with a k_{cat} of 255 s^{-1})^{70,71}. Therefore, we reason that the reaction mechanism itself should allow evolving lactyl-CoA mutase towards even higher turnovers than those observed for the native reaction of Hcm^{29,32,33}.*

For future applications in a cell-free context, optimization of the stability and oxygen-tolerance of Lcm may represent another important target, since inactivation of B₁₂-dependent mutases by molecular oxygen is well-described, especially in vitro⁷¹⁻⁷⁶. We emphasize that applications of Lcm in vivo allow the co-expression of chaperones (MeaB)^{71,77}, salvage of damaged coenzyme B₁₂⁷⁸ and improved protection of the coenzyme from photolysis and oxidative damage in the reductive intracellular environment. Applications in vivo thus likely present a more promising environment for future implementations of Lcm compared to in vitro systems.

In summary, Lcm and the Lcm module open new possibilities for metabolic engineering and might – upon further improvement in the future – become valuable additions to ongoing efforts of creating new C1-assimilating metabolic pathways for a sustainable bioeconomy.”

Despite many improvements and great potential, the Complexity of the System can be a bottleneck. The pathway involves multiple steps and enzymes, which could complicate the optimization and scaling-up processes.

We agree with the reviewer, the system is quite complex for an *in vitro* system. As noted above, we expanded the Discussion by adding corresponding considerations and emphasize that using the Lcm module *in vivo* currently shows greater promise than cell-free applications.

Some points will need to be improved in future research and should be mentioned in the text, like Enhance Enzyme Stability: Further research could focus on improving the stability of Lcm variants, particularly for those that show high catalytic efficiency but poor stability; Streamline the Pathway: Simplifying the pathway by reducing the number of steps or engineering multifunctional enzymes could improve its efficiency and feasibility for industrial applications and Optimizing the overall pathway to increase pyruvate yield and reduce by-product formation or substrate inhibition could make the process more economically viable.

In summary, while the study presents a promising new pathway for CO₂ assimilation and pyruvate production, further work is needed to address the limitations related to catalytic efficiency, enzyme stability, and system complexity. Additionally, the pathway's long-term practicality and economic viability in industrial applications must be thoroughly evaluated.

We agree with the reviewer that the manuscript would benefit from further discussing these limitations. We thus elaborated on some of these points in the discussion section as noted above.

Lanes 102-108 - The Lcm module outperforms other pathways with only 7 and 9 reactions for converting acetyl-CoA to pyruvate and oxaloacetate, respectively. Alternative routes lack experimental validation and suffer from oxygen-sensitive, inefficient biocatalysts. Some quantitative data demonstrating the Lcm module's efficiency would highlight its biotechnological advantages. Predictive analysis could further establish its superiority over existing alternatives.

Following this suggestion, we now provide an additional analysis of the thermodynamic favorability and resource requirement of all discussed acetyl-CoA assimilation routes at physiological conditions (see also answer to reviewer #1).

This analysis shows that, in terms of thermodynamic driving force, the Lcm module outperforms almost all other routes. The sole exception is the “partial 3HP-bicycle” with a slightly higher predicted driving force. However, this pathway is significantly longer than the Lcm module, requires twice as many enzyme steps and consumes 30% more ATP for conversion of acetyl-CoA into pyruvate. We emphasize that the thermodynamic favorability of the Lcm module is not tied to an increased resource cost (i.e. ATP and reducing equivalents), which is now summarized for the reader in Supplementary Figures 1, 2 and Supplementary Data 1. Overall, this data further supports our claims in respect to the efficiency of the Lcm module. We now refer to this new data in the main text as follows:

“The Lcm module outperforms most other acetyl-CoA assimilation routes in terms of thermodynamic favorability (max-min driving force, MDF) and ATP requirement (Supplementary Data 1, Supplementary Fig. 2).”

Line 117- ... *Kyrpidia tusciae* and *Bacillus massiliosenegalensis* (also referred to as *Robertmurray massiliosenegalensis*). Reference?

We thank the reviewer for pointing this out, we now added the relevant references to this sentence.

Discussion:

The provided text functions more as a conclusion than a discussion. It summarizes the study's achievements, highlighting the establishment of the Lcm module, its efficiency improvements, and its potential applications in synthetic biology. The text reflects on the outcomes and implications of the research, offering a succinct overview of the work's significance and future utility, which are characteristics typically found in a conclusion section. A discussion section would delve deeper into analyzing the results, comparing them with existing literature, and exploring the broader implications and limitations of the study.

Some Unresolved Questions could be briefly discussed in a proper discussion section.
-How stable is the engineered pathway in long-term cultures, and does it maintain efficiency under varying environmental and operational conditions?

-What are the potential economic and environmental impacts of scaling up this pathway for industrial use, especially regarding cost-effectiveness and carbon footprint reduction?

We thank the reviewer for this suggestion and agree that such questions will be important to consider. However, we believe that a detailed evaluation of the pathway performance under industrial scaled-up conditions or a techno-economic analysis comparing this pathway to other options for acetyl-CoA conversion are too early and outside the scope of this study. We aim to address these topics in the future.

-How well can the Lcm module be integrated into different host organisms, especially those used in industrial bioprocessing, without affecting their growth and viability?

We anticipate that the Lcm module can be integrated very well with *E. coli* metabolism and should not negatively impact growth, if expressed at suitable levels. Indeed, we found no indication of reduced viability due to Lcm enzymes during our *in vivo* implementations. However, we would argue that discussing predictions on pathway compatibility with various hosts would remain quite vague and speculative, even for metabolic networks as well-understood as that of *E. coli*. We propose that compatibility of Lcm module enzymes to the host metabolism of various industrially used microbes should ideally be tested experimentally in the future, providing more reliable data than theoretical considerations.

I recommend enhancing the discussion throughout the text, as the current discussion section resembles more of a conclusion. It would benefit from a more detailed analysis, comparing the findings with existing literature and exploring the broader implications and limitations of the study, as opposed to merely summarizing the outcomes and potential applications.

We thank the reviewer for this constructive criticism and now significantly expanded the Discussion section, including (i) a comparisons of the Lcm module to previous literature (e.g. bioproduction of 3-hydroxypropionate and compatibility to native host metabolism; carbon-conserving metabolic pathways; bioproduction from two-carbon feedstocks such as acetate and ethanol); (ii) added current limitations and emphasized the need to improve the module for such applications; (iii) discuss the potential for further improvement based on the activities of related enzymes.

The reference list requires meticulous verification to rectify missing data and format discrepancies, such as the improperly formatted reference 40.

We reviewed and corrected all entries in the reference list, including reference #40.

Point-by-point response

Reviewer #1 (Remarks to the Author):

The manuscript has been thoroughly revised and my main points of criticism have been addressed appropriately. From my side, it is almost acceptable for publication. However, there are still some little issues that should be checked.

We thank the reviewer for the kind words and thorough examination of our manuscript, which we perceived as very helpful in improving the work. We addressed the remaining points as described further below.

L147 “None of the mutases showed detectable activity with D-lactyl-CoA, which is in line with previous results demonstrating stereospecificity of these enzymes with their native substrate. 30,38”

Still, this might need some revision, as it not “in line” with the stereospecificity of the Aquincola enzyme. Yes, references 30 and 38 deal with the mutases preferring interconversion of R-3-hydroxybutyryl-CoA to 2-hydroxyisobutyryl-CoA. From this, I would also deduce preference for L-lactyl-CoA. However, the enzyme from Aquincola clearly prefers S-3-hydroxybutyryl-CoA, as the hydroxyl group is interacting with D117 via H-bonds (see PDB 4R3U, reference 31). Due to this, D-lactyl-CoA (R-lactyl-CoA) should be clearly favored.

We agree with the reviewer that the previously studied native activities do not directly allow deducing a consistent stereospecificity with lactyl-CoA for all tested enzyme candidates. To avoid misunderstandings, we chose to shorten this sentence to simply state our observation that none of the tested mutases showed detectable activity with D-lactyl-CoA:

“None of the mutases showed detectable activity with D-lactyl-CoA.”

L170 “The enzyme activity with L-lactyl-CoA was even slower ($k_{cat} = 0.03 \pm 0.01 \text{ s}^{-1}$), while the K_m value for L-lactyl-CoA ($0.14 \pm 0.09 \text{ mM}$) was higher than for the native substrate 2-hydroxyisobutyryl-CoA (Fig. 2C, D).” Yes, the rate is lower (10 times) and the K_m is higher (also 10 times). That means that both parameters (k_{cat} and K_m) are worse than with the native substrates = the efficiency is 100 times lower. Therefore, maybe, the use of “while” is somewhat misleading (when understood in the sense of “whereas” and indicating a contrast).

We thank the reviewer for pointing this out. We now exchanged the “while” by “and” to clarify the sentence:

“The enzyme activity with L-lactyl-CoA was even slower ($k_{cat} = 0.03 \pm 0.01 \text{ s}^{-1}$), and the K_m value for L-lactyl-CoA ($0.14 \pm 0.09 \text{ mM}$) was higher than for the native substrate 2-hydroxyisobutyryl-CoA (Fig. 2C, D).”

“ k_{cat} ”: use italic type throughout (e.g. L503).

We italicized k_{cat} throughout the manuscript.

Supplementary Figure 4:

“via the LCM over time (see panel C).” It is panel D now.

Thank you, we corrected the mistake.

“wild-type” (e.g. L90) versus “wild type” (e.g. L286) versus “wildtype” (e.g. L90Supplementary Figure 14).

Obviously, there are still some inconsistencies of writing in the manuscript.

We double-checked the manuscript to ensure we consistently use only the two correct spellings: “wild-type” when used as an adjective (e.g. “wild-type enzyme” in line 95) and “wild type” when used as a noun (e.g. “compared to the wild type” in line 409).

Reviewer #2 (Remarks to the Author):

Authors carefully addressed all the comments with more data and discussions, which further clarify the advances the authors have made in this manuscript. Now this reviewer recommends publication.

We thank the reviewer for the positive feedback and are grateful for the constructive criticism, which improved the manuscript.

Reviewer #3 (Remarks to the Author):

The study introduces a new CO₂-assimilating pathway—the Lcm module—which efficiently transforms acetyl-CoA into pyruvate without carbon loss, offering a faster, oxygen-tolerant alternative to traditional methods. The authors have done an excellent job revising the manuscript, thoughtfully incorporating both my suggestions and the recommendations of other reviewers. Their clear and convincing explanations for the changes made, or not made, leave me satisfied with their revisions.

We are grateful to the reviewer for the support and kind words, and the input which helped to improve the manuscript.